# Ca$^{2+}$ signals initiate at immobile IP$_3$ receptors adjacent to ER-plasma membrane junctions

Nagendra Babu Thillaiappan[1], Alap P. Chavda[1,2], Stephen C. Tovey[1,3], David L. Prole [1] & Colin W. Taylor[1]

IP$_3$ receptors (IP$_3$Rs) release Ca$^{2+}$ from the ER when they bind IP$_3$ and Ca$^{2+}$. The spatial organization of IP$_3$Rs determines both the propagation of Ca$^{2+}$ signals between IP$_3$Rs and the selective regulation of cellular responses. Here we use gene editing to fluorescently tag endogenous IP$_3$Rs, and super-resolution microscopy to determine the geography of IP$_3$Rs and Ca$^{2+}$ signals within living cells. We show that native IP$_3$Rs cluster within ER membranes. Most IP$_3$R clusters are mobile, moved by diffusion and microtubule motors. Ca$^{2+}$ signals are generated by a small population of immobile IP$_3$Rs. These IP$_3$Rs are licensed to respond, but they do not readily mix with mobile IP$_3$Rs. The licensed IP$_3$Rs reside alongside ER-plasma membrane junctions where STIM1, which regulates store-operated Ca$^{2+}$ entry, accumulates after depletion of Ca$^{2+}$ stores. IP$_3$Rs tethered close to ER-plasma membrane junctions are licensed to respond and optimally placed to be activated by endogenous IP$_3$ and to regulate Ca$^{2+}$ entry.

[1] Department of Pharmacology, University of Cambridge, Cambridge CB2 1PD, UK. [2] Present address: Cactus Communications Pvt Ltd, 510 Shalimar Morya Park, Andheri (West), Mumbai 400053, India. [3] Present address: Cairn Research Ltd, Graveney Road, Faversham, Kent ME13 8UP, UK. Correspondence and requests for materials should be addressed to D.L.P. (email: dp350@cam.ac.uk) or to C.W.T. (email: cwt1000@cam.ac.uk)

nositol 1,4,5-trisphosphate receptors (IP$_3$Rs) are intracellular Ca$^{2+}$ channels that are expressed in most animal cells and release Ca$^{2+}$ from the endoplasmic reticulum (ER) in response to the many signals that stimulate IP$_3$ formation. IP$_3$Rs are molecular coincidence detectors[1] because they open and release Ca$^{2+}$ only when they bind both IP$_3$ and Ca$^{2+}$. This dual regulation, where IP$_3$ binding primes IP$_3$Rs to respond to Ca$^{2+}$, allows IP$_3$Rs to respond to Ca$^{2+}$ from other channels[2] and to propagate Ca$^{2+}$ signals regeneratively between IP$_3$Rs[3]. The latter generates a hierarchy of Ca$^{2+}$ release events that can be detected using total internal reflection fluorescence microscopy (TIRFM). These 'optical patch-clamp' measurements[4] reveal tiny cytosolic Ca$^{2+}$ signals that reflect the openings of single IP$_3$Rs. As IP$_3$ concentrations increase, Ca$^{2+}$ released from an active IP$_3$R ignites the activity of its IP$_3$-bound neighbours to give larger local Ca$^{2+}$ signals ('Ca$^{2+}$ puffs'), and then Ca$^{2+}$ waves that spread across the cell. These global Ca$^{2+}$ waves become more frequent as IP$_3$ concentrations increase further[5]. The spatiotemporal complexity of IP$_3$-evoked Ca$^{2+}$ signals allows increases in cytosolic Ca$^{2+}$ concentration to selectively regulate diverse processes, including development, migration, metabolism and neuronal functions[6].

IP$_3$-evoked Ca$^{2+}$ release also links extracellular stimuli to store-operated Ca$^{2+}$ entry (SOCE)[7], which is activated when the ER loses Ca$^{2+}$. The core components of most SOCE pathways are the ER Ca$^{2+}$ sensor, stromal interaction molecule-1 (STIM1) and the

plasma membrane (PM) Ca$^{2+}$ channel, Orai[8]. Loss of Ca$^{2+}$ from the luminal EF hands of STIM1 causes STIM1 to oligomerize and expose cytosolic domains that bind to phosphatidylinositol 4,5-bisphosphate (PIP$_2$) and Orai. Binding of STIM1 to PIP$_2$ in the PM traps STIM1 at stable ER-PM junctions, where the two membranes are closely apposed[9–12]. Tethered STIM1 then captures Orai and stimulates opening of its Ca$^{2+}$-selective pore. SOCE is activated only after substantial loss of Ca$^{2+}$ from the ER[8,13], yet the ER must maintain other Ca$^{2+}$-dependent functions, notably protein synthesis and maturation, alongside its contributions to Ca$^{2+}$ signalling. Such conflicting demands suggest a need to segregate different functions within the ER[14].

Many studies suggest that all three IP$_3$R subtypes are mobile within ER membranes[15–17], and that signals arising from stimulation of phospholipase C (PLC) promote IP$_3$R clustering[18–24]. Because most of these studies relied on over-expression of tagged IP$_3$Rs, they may not faithfully report the behaviour of endogenous IP$_3$Rs. Furthermore, the conclusion that most IP$_3$Rs are mobile contrasts with evidence that IP$_3$-evoked Ca$^{2+}$ signals often initiate repeatedly, and over protracted periods of stimulation, from the same intracellular sites[25–27]. The lack of any satisfactory explanation for this apparently paradoxical mobility of IP$_3$Rs and immobility of Ca$^{2+}$ release sites highlights the need to understand how the subcellular geography of IP$_3$Rs and Ca$^{2+}$ signals are related.

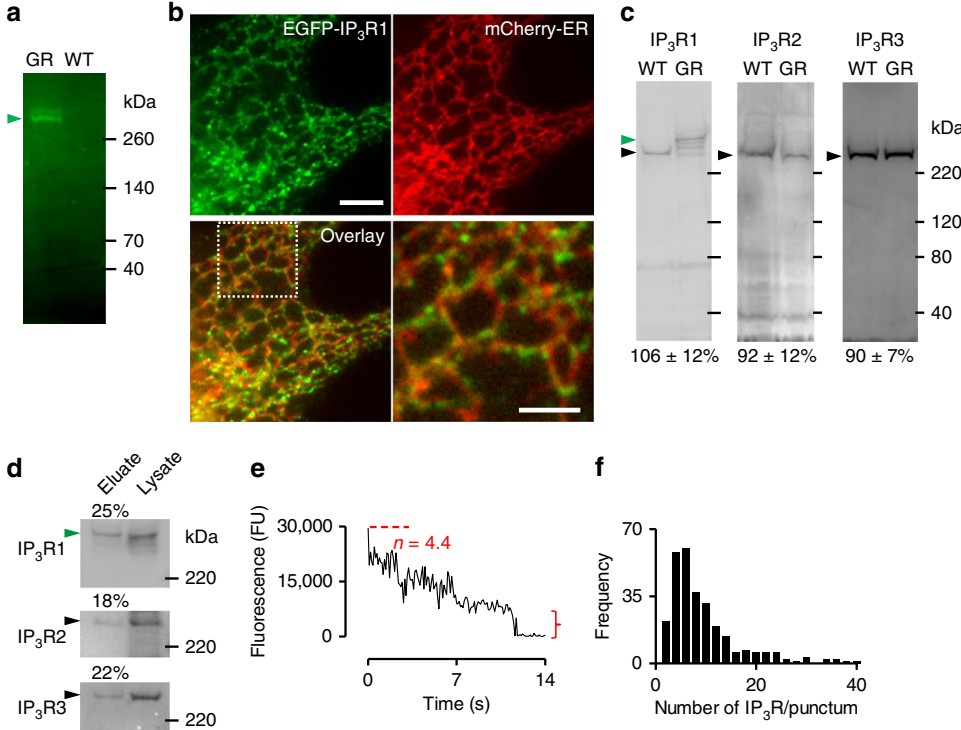

**Fig. 1** Endogenous IP$_3$R1s form puncta. **a** In-gel fluorescence of lysates from EGFP-IP$_3$R1 HeLa cells (GR) and control (WT) cells demonstrates that the only fluorescence is associated with EGFP-IP$_3$R1 (green arrow). Results typical of four gels. Positions of selected $M_r$ markers (kDa) are shown (**a**, **c**, **d**). **b** TIRFM images of EGFP-IP$_3$R1 HeLa cells showing a marker for the ER lumen (mCherry-ER). The merged image and an enlargement of the boxed area show co-localization of EGFP-IP$_3$R1 with mCherry-ER (Pearson's coefficient with Costes' automatic threshold = 0.93 ± 0.02; Costes $P$ value = 1.00, $n = 4$ cells). Scale bar = 5 μm (2 μm for enlargement). **c** Western blots (WBs) for IP$_3$R1-3 show expression of tagged (green arrow, ~290 kDa) and untagged (black arrow, ~260 kDa) IP$_3$R1 in GR and WT cells, respectively. Expression of IP$_3$R subtypes in GR cells is shown relative to control (WT) cells (%, mean ± SD, $n = 3$ for IP$_3$R2 and IP$_3$R3, $n = 4$ for IP$_3$R1). Comparisons of band intensities using paired Student's $t$-tests indicated no significant differences between WT and EGFP-IP$_3$R1 cells. **d** WB (IP$_3$R1-3 antibodies) from lysates of EGFP-IP$_3$R1 HeLa cells after immunoprecipitation with GFP-Trap. Eluate lanes were loaded with sample equivalent to 1.5 times the amounts loaded in the lysate lanes. Numbers show % of each subtype detected in the pull-down ($n = 2$). **e** Photobleaching of a punctum showing the final bleaching step (bracket) and the initial fluorescence (dashed line) used to calculate the total number of fluorophores ($n$). FU, fluorescence units. **f** Single-step photobleaching results (284 puncta from five cells, Supplementary Fig. 5) were used to calculate the number of tetrameric IP$_3$Rs per punctum (8.4 ± 7)

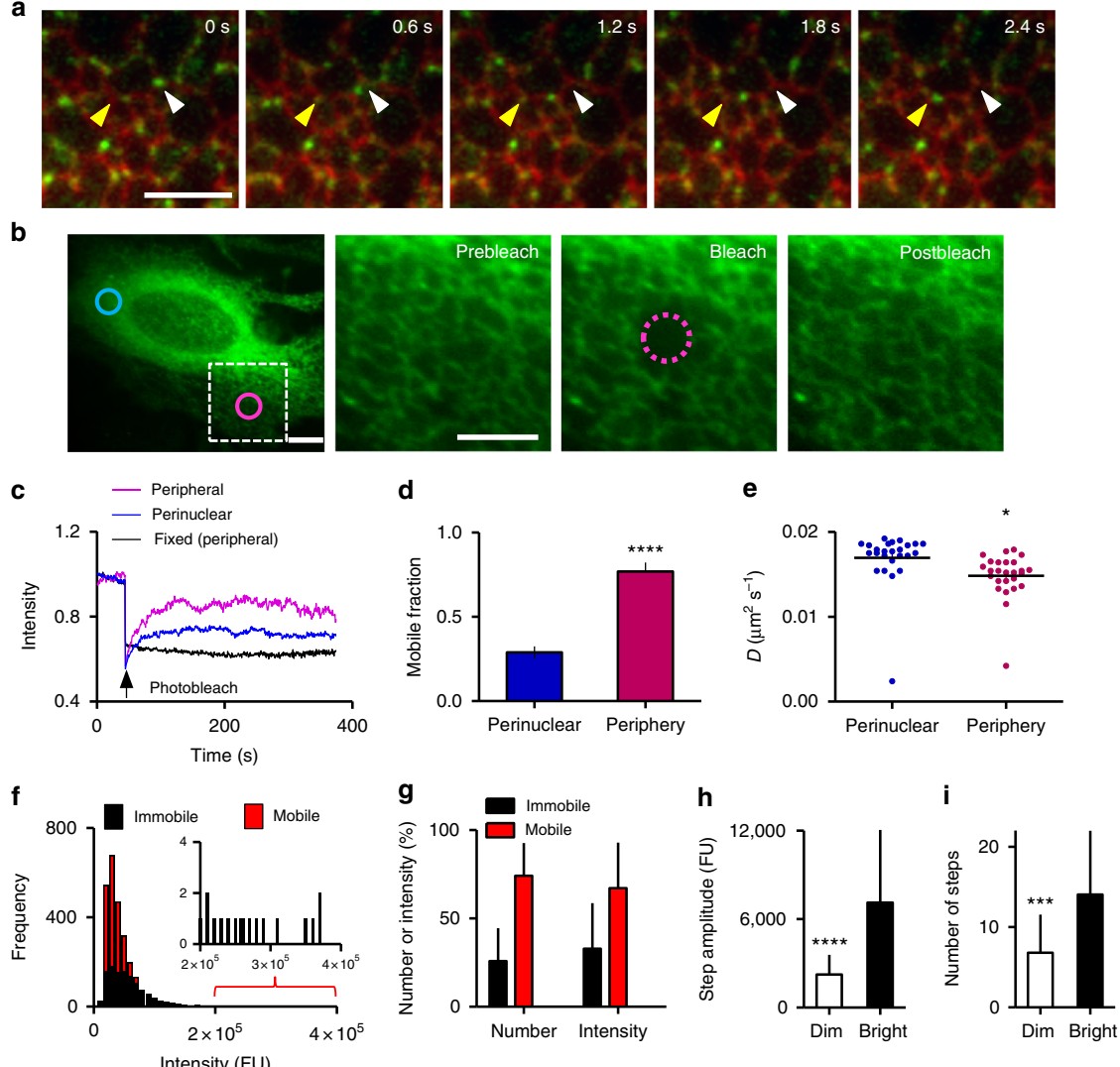

**Fig. 2** IP$_3$Rs form mobile and immobile puncta. **a** Time-lapse TIRFM images (0.6-s intervals) of EGFP-IP$_3$R1 in cells expressing mCherry-ER. Track of a single particle, with the first and last positions shown by white and yellow arrows, respectively. Scale bar = 5 μm. **b** Representative epifluorescence image of an EGFP-IP$_3$R1 HeLa cell with perinuclear (blue) and peripheral (magenta) regions highlighted for FRAP analysis (circular bleached area, radius = 1.84 μm). The boxed area is enlarged to show pre- and post-bleach (after 120 s) images of the peripheral region. Scale bars = 5 μm. **c** Normalized fluorescence intensities recorded from peripheral or perinuclear regions in a typical FRAP experiment with live and fixed EGFP-IP$_3$R1 HeLa cells. **d, e** Summary results show mobile fractions ($M_f$, mean ± SEM) (**d**) and diffusion coefficients ($D$, mean and all values) (**e**) for perinuclear (25 cells) and peripheral regions (26 cells). ****$P < 0.0001$, *$P < 0.05$, two-tailed Student's $t$-test. **f** Distribution of fluorescence intensities for individual mobile and immobile puncta. Inset shows distribution for the brightest immobile puncta. **g** Relative numbers of mobile and immobile puncta, and distribution of fluorescence between them (%). Results (**f, g**) are from time-lapse TIRFM images of 10 cells (mean ± SD). **h, i** Analysis of bleaching steps of brightest and dimmest puncta in fixed cells was used to determine the step amplitude (**h**) and number of steps (**i**) for each punctum (Supplementary Fig. 8). Mean ± SD, with 23–25 puncta analysed in each of two cells. ***$P < 0.001$, ****$P < 0.0001$, Student's $t$-test

Here, we use gene editing to tag endogenous IP$_3$R1 of human cells with monomeric enhanced green fluorescent protein (EGFP). The EGFP-IP$_3$R1s are functional and allowed high-resolution measurements of both IP$_3$R distribution and the Ca$^{2+}$ signals evoked by IP$_3$. We show that a small pool of immobile IP$_3$Rs tethered close to ER-PM junctions are 'licensed' to respond to IP$_3$. These IP$_3$Rs are optimally placed to respond to IP$_3$ delivered from PLC and to regulate SOCE.

## Results

**Endogenous IP$_3$Rs are clustered.** IP$_3$R1 is the major IP$_3$R subtype in HeLa cells[28]. We used transcription activator-like effector nucleases (TALENs)[29] to tag all endogenous IP$_3$R1 of HeLa cells with monomeric EGFP (Supplementary Figs. 1, 2). We

chose EGFP for tagging because we had previously tagged endogenous IP$_3$R1 with mCherry and then detected substantial fluorescence in lysosomes, wherein mCherry but not EGFP is relatively stable[30]. There were no such problems with EGFP-IP$_3$R1 (Supplementary Fig. 3). Within the gene-edited EGFP-IP$_3$R1 HeLa cells, EGFP-IP$_3$R1 was functional and there was no off-target tagging of other proteins (Fig. 1a and Supplementary Figs. 1–3). In live cells, EGFP-IP$_3$R1 was expressed in ER membranes and formed discrete puncta within them (Fig. 1b and Supplementary Fig. 4). Most cells express more than one of the three IP$_3$R subtypes, and the subunits can form heterotetrameric channels. HeLa cells express IP$_3$R1 (61%), IP$_3$R2 (9%) and IP$_3$R3 (30%)[28]. This pattern of expression was unaffected by tagging endogenous IP$_3$R1 with EGFP (Fig. 1c), and

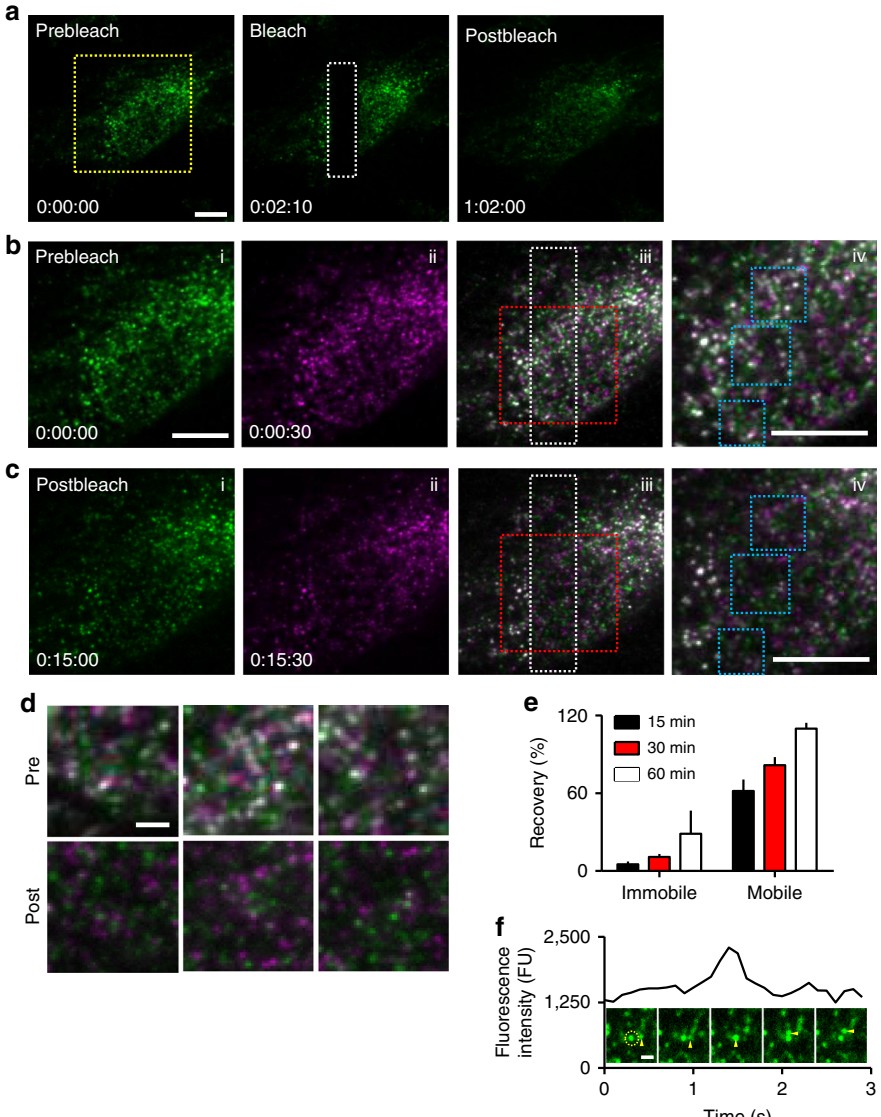

**Fig. 3** IP$_3$Rs within mobile puncta do not exchange with immobile puncta. **a** TIRFM images of FRAP experiment show images before, immediately after and 60 min after bleaching. Bleached area shown by white rectangle. Times shown as h:min:s. **b**, **c** Enlarged images of yellow boxed area in **a**. Immobile puncta were identified by overlaying two frames (30 s apart) using pseudocolours for each (green and magenta), such that immobile puncta appear white in the overlay (iii, iv) (Supplementary Fig. 7). Images were captured before photobleaching (**b**) and after recovery for 15 min (**c**). Enlarged areas (red boxes in iii) are shown in (iv). Scale bars (**a**–**c**) = 10 μm. **d** Enlargements (blue boxes in **b** iv and **c** iv) show images before and 15 min after bleaching. Scale bar = 2 μm. Abundant green and magenta puncta in the post-bleach images (**c**, and lower in **d**) indicate rapid exchange of mobile EGFP-IP$_3$R, while the scarcity of white puncta suggests very slow exchange of immobile EGFP-IP$_3$R1. **e** Summary results show fluorescence recovery ($F/F_0$, %, mean ± SEM) from three cells (26 ± 15 immobile puncta per cell were identified in the initial images). **f** Example trace from a region centred on an immobile punctum (circled in first image) shows stepwise increase and decrease in fluorescence intensity as a mobile punctum (arrows) moves through the immobile punctum (Supplementary Movie 2). Scale bar = 1 μm

co-immunoprecipitation confirmed that EGFP-IP$_3$R1 co-assembles with endogenous IP$_3$R2 and IP$_3$R3 subunits (Fig. 1d). Assuming that IP$_3$R subunits assemble randomly into tetramers, which may be an over-simplification[31], but it is consistent with our immunoprecipitation results (Fig. 1d), we expect about 98% of all tetrameric IP$_3$Rs to include at least one IP$_3$R1 subunit. Since all IP$_3$R1 are tagged with EGFP (Fig. 1c) and under optimal conditions about 80% of expressed EGFP is fluorescent[32], about 93% of all tetrameric IP$_3$Rs in EGFP-IP$_3$R1 HeLa cells should include at least one fluorescent subunit, suggesting that most IP$_3$Rs are visualized in our gene-edited cells.

TIRFM and single-step photobleaching[32] of fixed EGFP-IP$_3$R1 HeLa cells were used to determine the number of fluorophores associated with each punctum, and thereby the mean number of tetrameric IP$_3$Rs per punctum (8.4, range from 1 to ~40) (Fig. 1e, f and Supplementary Fig. 5). Since TIRFM, which is required for the photobleaching analyses, only detects EGFP-IP$_3$R1 within about 140 nm of the PM, we used spinning disc confocal microscopy to examine the distribution of EGFP-IP$_3$R1 in the rest of the cell. EGFP-IP$_3$R1 puncta, with features similar to those observed by TIRFM, were present throughout the cell (Supplementary Fig. 6). We conclude that many near-PM IP$_3$Rs, and probably most IP$_3$Rs in the cell, are assembled into small clusters, suggesting that these clusters are the elementary structural units of IP$_3$R signalling. This is consistent with our observation that we rarely observed splitting of puncta (Supplementary Movies 1–3).

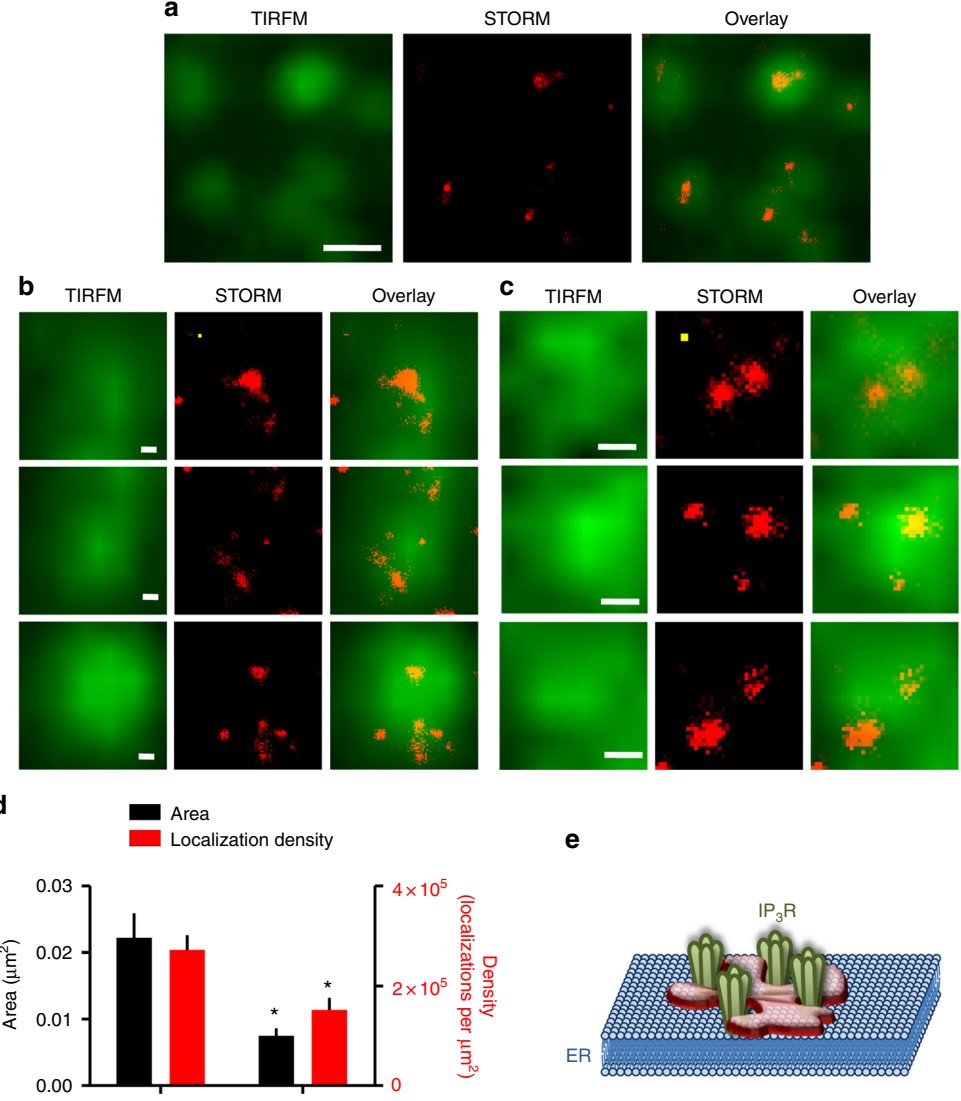

**Fig. 4** EGFP-IP$_3$Rs are diffusely distributed within puncta. **a** Examples of TIRFM and subsequent STORM images of EGFP-IP$_3$R1 puncta. Scale bar = 0.5 μm. **b**, **c** Before STORM, mobile and immobile puncta were identified using TIRFM and by overlaying pseudocoloured images collected at 10-s intervals (Supplementary Fig. 7). Examples of TIRFM and STORM images show immobile (**b**) and mobile puncta (**c**). Overlays show that EGFP-IP$_3$R1s are diffusively distributed within mobile and immobile puncta. Scale bars = 100 nm. The yellow squares show approximate dimensions of an IP$_3$R (20 nm × 20 nm). **d** Analysis of STORM images shows the area enclosing all EGFP-IP$_3$R1s within each punctum and the density of localization events within the punctum for mobile and immobile puncta (mean ± SD for 4–6 puncta). *$P < 0.05$, Student's $t$-test, relative to immobile puncta. Results show that mobile puncta are smaller and contain fewer IP$_3$Rs than immobile puncta. **e** Many IP$_3$Rs within puncta are too diffusely distributed to allow direct interactions between them. Although we used a non-oligomerizing EGFP to tag IP$_3$R1 and minimize potential artefacts, the diffuse spacing of some EGFP-IP$_3$R1s confirms that puncta are not formed by interactions between EGFP

**Mobile and immobile populations of IP$_3$Rs.** Most EGFP-IP$_3$R1 puncta were mobile (74 ± 18%, mean ± SEM, $n = 10$ cells), while other puncta remained immobile for many minutes (Fig. 2a and Supplementary Movie 1). Fluorescence recovery after photobleaching (FRAP) confirmed the presence of mobile and immobile EGFP-IP$_3$R1s (Fig. 2b–e). Within the cell periphery, the mobile fraction ($M_f$) was significantly larger than in perinuclear regions (Fig. 2d). The half-times for recovery in peripheral and perinuclear regions ($t_{1/2} = 62 ± 6$ s, $n = 26$ cells and 61 ± 12 s, $n = 25$ cells, respectively) and the calculated diffusion coefficients ($D$) were similar in each region (Fig. 2e). We present estimates of $D$ to allow comparisons with published data[15–17], but subsequent experiments demonstrate that only some mobile IP$_3$R puncta move by diffusion alone. A better (but similar) estimate of

$D$ is provided by our analysis of puncta shown to move only by diffusion ($D = 0.0308 ± 0.002$ μm$^2$ s$^{-1}$) (see Supplementary Fig. 10a, e).

The TIRFM analyses of mobility, which exclude perinuclear regions but include the periphery, suggest that 74% of puncta are mobile. This matches the $M_f$ determined by FRAP in the cell periphery (0.77). The similarity might suggest that IP$_3$Rs within immobile puncta do not exchange with mobile IP$_3$Rs. We tested this directly with a FRAP experiment in which recoveries of fluorescence within mobile and immobile puncta were independently quantified (Fig. 3a–e and Supplementary Fig. 7). The fluorescence of immobile puncta recovered very slowly: after 15 min, fluorescence from mobile puncta had recovered to 62 ± 9% ($n = 3$) of its initial value, while that from immobile

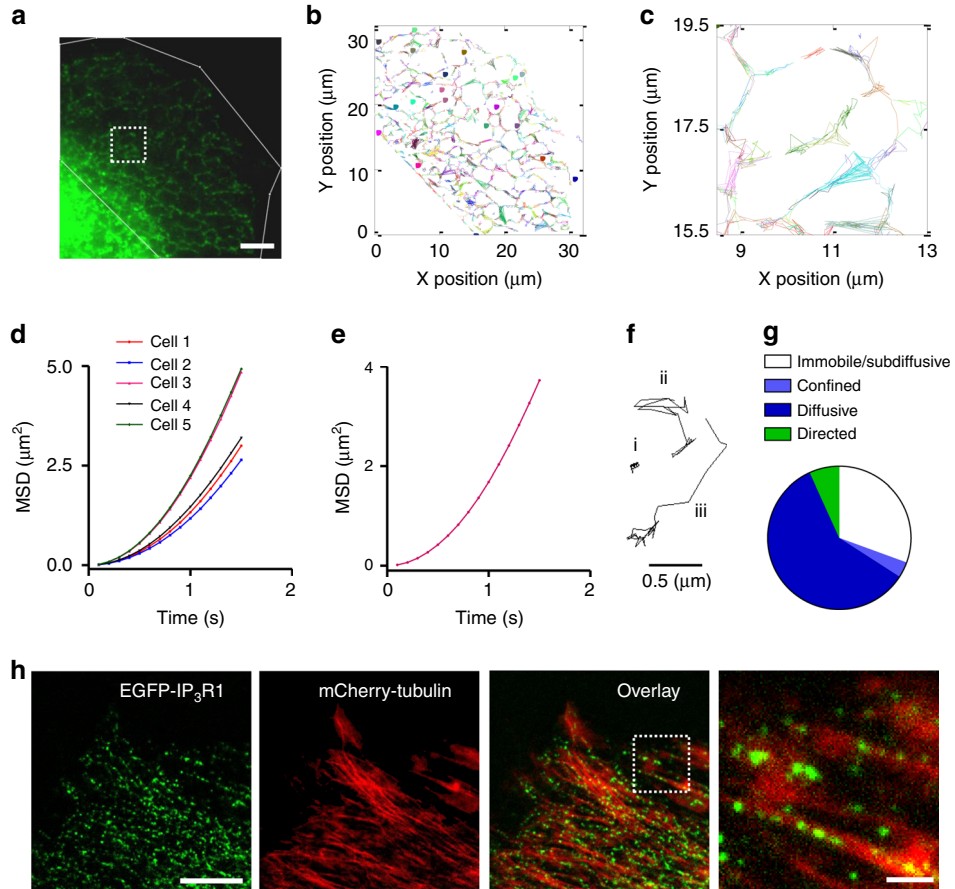

**Fig. 5** IP$_3$Rs move by diffusion and by directed motion along microtubules. **a** TIRFM image used for single-particle tracking shows characteristic reticular ER. Scale bar = 5 μm. **b** Trajectories of all puncta from the region defined by white lines in **a**. Colours indicate different trajectories. **c** Enlargement showing trajectories within the boxed area in **a**. **d** Average mean squared displacement (MSD) for all puncta within each cell plotted against time. Results are from five cells, with 225–926 puncta analysed in each. **e** Pooled results from five cells (2,698 tracks). **f** Examples of single-particle trajectories (collected at 100-ms intervals) for an immobile punctum (i) or mobile puncta moving by diffusion (ii) or directionally (iii). **g** Distribution of EGFP-IP$_3$R1 puncta between mobility states. Trajectories of 1,509 puncta from five cells were analyzed by TraJClassifier (Supplementary Fig. 10). **h** Wide-field images, typical of nine cells, show EGFP-IP$_3$R1 puncta, microtubules (mCherry-tubulin) and their co-localization. Final panel shows an enlargement of the boxed area in the overlay. Scale bar = 10 μm (2 μm in enlargement)

puncta had recovered by only 5.1 ± 2.2% (Fig. 3e). We also occasionally observed examples of mobile puncta seemingly passing unperturbed through an immobile punctum (Fig. 3f and Supplementary Movie 2). We conclude that there is negligible exchange of IP$_3$Rs over many minutes between mobile and immobile puncta.

**IP$_3$Rs are diffusely distributed within puncta.** Most mobile and immobile puncta had similar distributions of fluorescence intensity, but a small fraction of immobile puncta (1.6%) were much brighter (about three- to eightfold brighter than average) (Fig. 2f, g). The difference might be due to bright puncta including more EGFP-IP$_3$R1s, or to them being closer to the PM (and so brighter in TIRFM)[33]. The amplitude of the single-step bleaching events and the number of bleaching steps were each about twofold larger for the brightest puncta (Fig. 2h, i and Supplementary Fig. 8). This suggests that the brightest immobile puncta, which include about 3% of the fluorescence detected within puncta, are both closer to the PM and include more EGFP-IP$_3$R1s than other puncta.

We used stochastic optical reconstruction microscopy (STORM) to resolve the distribution of EGFP-IP$_3$R1s within puncta. The fixation required for STORM had no effect on the size or distribution of the puncta observed in live cells

(Supplementary Fig. 9). By recording from live cells before fixation, we could identify mobile and immobile puncta in the subsequent STORM images (Fig. 4a). The resolution of these images (~45 nm) is not sufficient to distinguish single IP$_3$Rs from very small clusters (of up to about four IP$_3$Rs), but it is clear that the distribution of fluorophores within most immobile puncta was heterogeneous: some IP$_3$Rs were tightly packed, while others were more loosely distributed (Fig. 4b). The distances between some IP$_3$Rs within a punctum (up to about 200 nm) were too large for the punctum to be assembled by direct interactions between IP$_3$Rs, as each tetrameric IP$_3$R has a diameter of only 20 nm[34]. Similar analyses of mobile puncta revealed that although IP$_3$Rs were more tightly packed than in immobile puncta, here too the spacing of IP$_3$Rs was often too large for direct interactions between IP$_3$Rs to mediate their association (Fig. 4c, d). We conclude that IP$_3$Rs are corralled within mobile and immobile puncta by mechanisms that do not require direct contact between IP$_3$Rs (Fig. 4e).

**IP$_3$Rs are moved by diffusion and microtubule motors.** We used TIRFM and single-particle tracking to quantify movements of EGFP-IP$_3$R1 puncta within the ER (Fig. 5a–c and Supplementary Fig. 10). Only the flattest regions of the cell, remote from the nucleus and typically comprising 30–50% of the cell area,

were amenable to these analyses. If all mobile $IP_3Rs$ moved randomly by diffusion, the relationship between time and mean squared displacement should be linear (Supplementary Fig. 10a, e), or sub-linear if diffusion is impeded by the labyrinthine architecture of the ER[35]. However, single-particle trajectories of 2,698 tracks from five cells demonstrate that the relationship is supra-linear (Fig. 5d, e), suggesting that at least some puncta move directionally. The trajectories of individual puncta revealed that some particles moved randomly, while others showed directional movement (Fig. 5f). These analyses identified distinct populations: immobile puncta ($31 \pm 4\%$ of puncta were immobile or sub-diffusive) and mobile puncta that moved by diffusion ($63 \pm 2\%$) or directionally ($7 \pm 2\%$) (Fig. 5g and Supplementary Fig. 10). Most puncta (96%) maintained a single form of motion throughout the $> 3$-s recording (see Methods).

We next addressed the mechanisms of directional movement. There was significant co-localization of EGFP-$IP_3R1$ with microtubules (Fig. 5h). The ER is itself dynamic, with microtubules contributing to its mobility directly and via kinesin-1 and cytoplasmic dynein motors[36]. However, $IP_3Rs$ were much more mobile than the ER (Supplementary Movie 3), demonstrating that $IP_3Rs$ move within the ER. Disrupting microtubule dynamics has been shown to slow $IP_3R$ movement, attenuate $IP_3$-evoked $Ca^{2+}$ signals and disrupt their spatial organization (see references in [24]), but many of these effects could be secondary to changes in ER structure. In EGFP-$IP_3R1$ HeLa cells, disrupting microtubule polymerization with nocodazole caused shrinkage of microtubules and slow retraction of the ER from the PM (Supplementary Fig. 11a, b), consistent with previous reports[37]. Nocodazole reduced the number of directional trajectories without affecting either the speed of the few that remained, or $D$ for the diffusing puncta (Supplementary Fig. 11c–e). Both the ER and $IP_3R3$ can associate with the growing plus-ends of microtubules via end-binding (EB) proteins[24], but there was no specific co-localization of mCherry-EB3 and directionally moving EGFP-$IP_3R1$ puncta (Supplementary Movie 4 and Supplementary Fig. 11f–h). Since EGFP-$IP_3R1$ puncta appear not to move by association with EB proteins, we considered a role for microtubule motors (Supplementary Fig. 12a).

GFP-Trap precipitated kinesin-1 (KIF5) from lysates of EGFP-$IP_3R1$ HeLa cells, and in the reciprocal analysis a kinesin-1 antibody pulled down $IP_3R1$. A dynein antibody also precipitated EGFP-$IP_3R1$ (Supplementary Fig. 12b–d). These results show that kinesin-1 and dynein interact with $IP_3R1$, directly or through additional proteins. We used rapamycin-mediated dimerization to trap kinesin-1 at the PM after it had moved there along microtubules[38]. PM trapping of kinesin-1 recruited EGFP-$IP_3R1$s, but not GFP-$ER_{memb}$ (a control ER membrane protein[36]), to the same PM sites, confirming that $IP_3R1$ can be conveyed along microtubules by kinesin-1 (Supplementary Movie 5 and Supplementary Fig. 13).

Expression of dominant-negative forms of kinesin heavy chain (KHC-CT) or cytoplasmic dynein (p50-mCherry)[36], or treatment with ciliobrevin D to inhibit dynein motors[39] reduced the number of puncta moving directionally, but the effects were not statistically significant (Supplementary Fig. 12e). Simultaneous inhibition of both motors significantly reduced both the number of directionally moving puncta and the speed of the few remaining directional puncta (Supplementary Fig. 12f, g). Evidence that kinesin and dynein contribute to directed movement of $IP_3Rs$ is consistent with the speed of directionally moving puncta ($0.324 \pm 0.040\,\mu m\,s^{-1}$; Supplementary Fig. 10d, h, i), which falls within the range of speeds reported for cargoes moved by mammalian kinesin[40] and dynein[41] motors.

We considered whether mobile $IP_3Rs$ might be held within membrane-bound vesicles separate from the ER[42]. This seems unlikely since there was no coincident concentration of a luminal protein (mCherry-ER) with moving EGFP-$IP_3R1$ puncta (Supplementary Movie 6 and Supplementary Fig. 14a), and directional EGFP-$IP_3R1$ trajectories overlaid the reticular ER defined by the trajectories of diffusing puncta (Supplementary Fig. 14b). We conclude that $IP_3Rs$ move within ER membranes by diffusion, or by directed motion that depends in part on kinesin and dynein motors.

**$Ca^{2+}$ puffs occur at immobile puncta close to the PM.** We used a red fluorescent $Ca^{2+}$ indicator and TIRFM to visualize local $Ca^{2+}$ release events in EGFP-$IP_3R1$ HeLa cells. Histamine evoked transient local $Ca^{2+}$ signals (Supplementary Movie 7 and Supplementary Fig. 15a). These '$Ca^{2+}$ puffs' were similar to those recorded in other cells[3]. Spinning disc confocal microscopy confirmed that almost all histamine-evoked local $Ca^{2+}$ signals occurred close to the PM. During matched recording intervals, we detected $69 \pm 20$ $Ca^{2+}$ puffs per cell ($n = 3$ cells) close to the basal PM, but only $1.3 \pm 1.2$ puffs in a plane across the middle of the cell and even these few events (four in total) were at the cell periphery close to the PM (see Supplementary Fig. 16c, d). The occurrence of most histamine-evoked $Ca^{2+}$ puffs close to the PM could result from targeted delivery of $IP_3$ from endogenous signalling pathways at the PM. However, photolysis of caged-$IP_3$, which uniformly releases $IP_3$ throughout the cell, evoked $Ca^{2+}$ puffs similar to those evoked by histamine (Figs 6a–e), and they too were concentrated near the PM, consistent with results from SH-SY5Y cells[43]. In matched recordings from HeLa cells responding to photolysis of caged-$IP_3$, we detected $103 \pm 30$ $Ca^{2+}$ puffs per cell ($n = 3$ cells) close to the basal PM, but only $7 \pm 2$ puffs per cell in a plane across the middle of the cell. The latter were invariably close to the PM (Supplementary Fig. 16). We conclude that $Ca^{2+}$ puffs, whether evoked by endogenous signalling pathways or by uniform delivery of $IP_3$ to the cytosol, are concentrated near the PM despite the presence of $IP_3R$ clusters throughout the cell (Supplementary Figs 6, 16d).

In the 30 s after photorelease of $IP_3$, TIRFM detected $295 \pm 20$ $Ca^{2+}$ puffs per cell ($n = 3$ cells) distributed across $75 \pm 13$ sites per cell, confirming that puffs often revisit the same site (Fig. 6d and Supplementary Movie 8). The response to histamine ($10\,\mu M$) was similar, with 125–307 puffs per cell distributed across 58–85 sites per cell (two cells) (Supplementary Movie 7). The occurrence of most $Ca^{2+}$ puffs at a limited number of sites is consistent with other reports[26,27], although our results suggest there are many more such sites (>70 per cell) than previously reported (typically 4–5 per cell)[3,44]. Most $Ca^{2+}$ puffs, whether evoked by histamine or photolysis of caged-$IP_3$, co-localized with immobile EGFP-$IP_3R1$ puncta (Fig. 6a–d and Supplementary Fig. 15). In the 30 s after photorelease of $IP_3$, $74 \pm 11\%$ of the 295 $Ca^{2+}$ puffs detected in three cells occurred at immobile puncta (Fig. 6f), and $34 \pm 8\%$ of the immobile puncta coincided with a $Ca^{2+}$ puff. In these analyses, where mobile and immobile $IP_3Rs$ were identified from images collected immediately before and after imaging of $Ca^{2+}$ signals for 30 s, we can identify most $Ca^{2+}$ puffs, but cannot define the locations of moving puncta during the $Ca^{2+}$ puffs (Fig. 6 and Supplementary Fig. 15). By collecting rapidly interleaved images of EGFP-$IP_3R1$ and $Ca^{2+}$ signals, we come much closer to relating the location of mobile puncta to $Ca^{2+}$ puffs, although we lose the opportunity to detect all $Ca^{2+}$ puffs. We note that the distances travelled by moving $IP_3R$ puncta (~200 nm) during the interleaved recordings (Supplementary Fig. 10) are so short that they would not affect our ability to identify spatially coincident $Ca^{2+}$ puffs and mobile $IP_3R$ puncta. Applying this approach to cells stimulated with histamine or

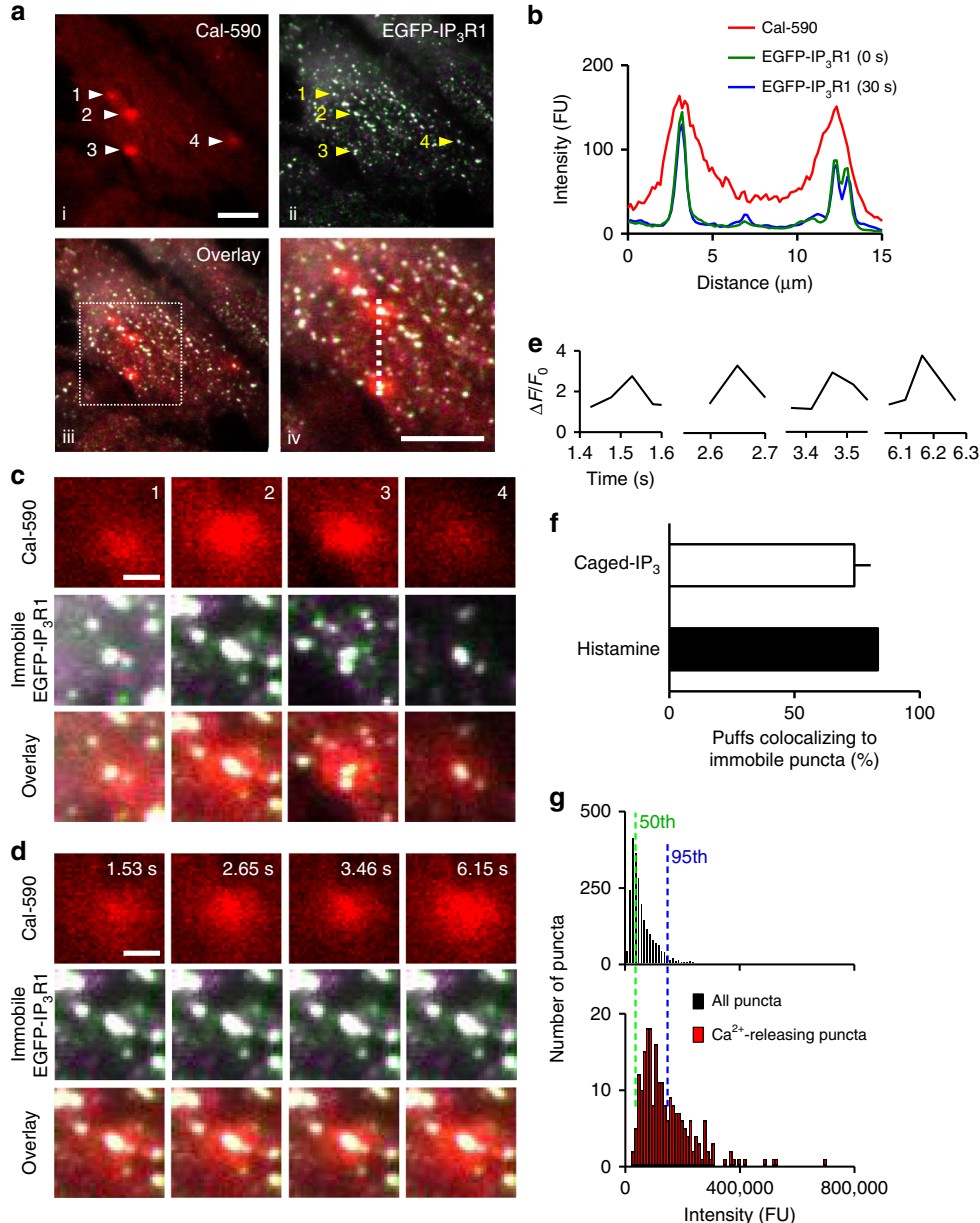

**Fig. 6** $Ca^{2+}$ puffs evoked by photolysis of caged-$IP_3$ occur at immobile $IP_3Rs$. **a** TIRFM image of a single cell loaded with EGTA and Cal-590 shows immobile $IP_3Rs$ (white, see Supplementary Fig. 7) and $Ca^{2+}$ puffs (red) evoked by photolysis of caged-$IP_3$. Boxed area of the overlay (iii) is shown enlarged in (iv). Scale bars = 10 μm. **b** Co-localization of $Ca^{2+}$ release and immobile $IP_3R$ puncta shown by their fluorescence intensity profiles along the dashed line in **a** iv. EGFP profiles were captured immediately before and ~30 s after recording Cal-590 fluorescence. **c** Enlargements of four puff sites highlighted in **a** show co-localization of each $Ca^{2+}$ release event with an immobile $IP_3R$ punctum. **d** Enlargements of site 2 (from **a**) show four successive puffs generated at the same immobile punctum. Scale bars = 2 μm (**c**, **d**). **e** Temporal profiles of Cal-590 fluorescence changes ($\Delta F/F_0$) show $Ca^{2+}$ puffs last ~200 ms. **f** Summary shows the fraction of $Ca^{2+}$ puffs evoked by histamine (432 puffs in two cells, mean ± range) or photorelease of $IP_3$ (871 puffs in three cells, mean ± SD) occurring at immobile $IP_3R$ puncta. Events and puncta were considered to occur at the same site if the centre of mass of the peak change in Cal-590 fluorescence was within 6 pixels (0.96 μm) of the punctum. **g** Comparison of fluorescence intensity distributions of all detected EGFP-$IP_3R1$ puncta, and puncta associated with $Ca^{2+}$ puffs evoked by photolysis of caged-$IP_3$. Results are from three cells. Dashed lines show 50th and 95th percentiles for the intensities of all puncta

photolysis of caged-$IP_3$, we confirmed that most $Ca^{2+}$ puffs co-localized with stable immobile $IP_3Rs$ (Supplementary Fig. 17).

Our results establish that $Ca^{2+}$ puffs are generated by immobile $IP_3Rs$ that are segregated from the more abundant pool of mobile $IP_3Rs$. Some immobile puncta are larger than mobile puncta (Fig. 2f, g), and larger clusters of $IP_3Rs$ have an increased probability of initiating $Ca^{2+}$ puffs[45]. However, this cannot be the only explanation for $Ca^{2+}$ puffs occurring almost exclusively at immobile puncta (Fig. 6f and Supplementary Fig. 17e). First,

responsive puncta lie close to the PM (Supplementary Fig. 16), which probably increases their brightness in TIRFM and so exaggerates (by ~2-fold) the estimated number of $IP_3Rs$ within them (Fig. 2h). Second, the number of immobile puncta shown to initiate $Ca^{2+}$ signals (34%, during a 30-s recording) is much larger than the number of unusually bright immobile puncta (only 1.6% were 3- to 8-fold brighter than the average punctum) (Fig. 2f). Third, although bright immobile puncta do initiate $Ca^{2+}$ signals, many of the puncta that released $Ca^{2+}$ had fluorescence

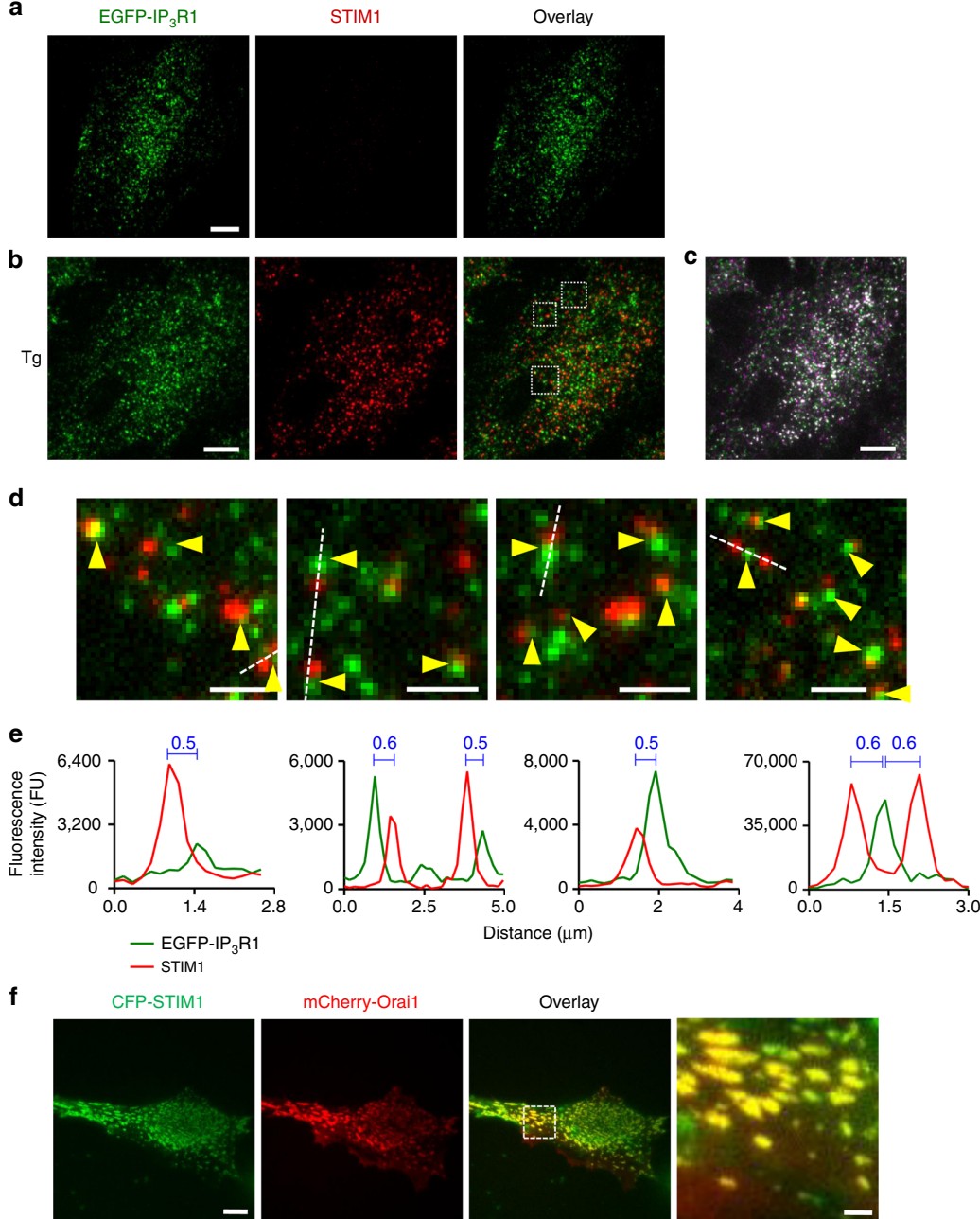

**Fig. 7** Depletion of ER $Ca^{2+}$ stores causes native STIM1 to accumulate at functional ER-PM junctions adjacent to immobile $IP_3R$ puncta. **a, b** Representative TIRFM images of EGFP-$IP_3R1$ HeLa cells fixed and immunostained for STIM1 before (**a**) or after treatment with thapsigargin (Tg, 1 μM, 15 min) to deplete the ER of $Ca^{2+}$ (**b**). Overlaid images of Tg-treated cells show no significant co-localization of STIM1 and $IP_3R$ puncta (Pearson's coefficient with Costes' automatic threshold: 0.331 ± 0.026, $n = 7$ cells). **c** Distribution of mobile (green and magenta) and immobile (white) $IP_3R$ puncta in Tg-treated cell. Scale bars (**a**–**c**) = 10 μm. **d** Enlargements of the boxed regions in **b** show that immobile $IP_3R$ puncta (identified before fixation (**c**), with all shown by arrowheads) abut STIM1 puncta without coinciding with them. Scale bars = 2 μm. **e** Fluorescence intensity profiles for EGFP-$IP_3R1$ and STIM1 across the lines shown in **d**. Distances (μm) between the peaks of the fluorescence intensity for STIM1 and immobile $IP_3R$ are shown. **f** Co-localization of CFP-STIM1 (pseudocoloured green) and mCherry-Orai1 (red) puncta in a Tg-treated HeLa cell. We used tagged proteins because available antibodies do not reliably detect endogenous Orai1. Scale bar = 10 μm (2 μm in enlargement)

intensities comparable to those of mobile puncta (Fig. 6g). We conclude that $Ca^{2+}$ puffs evoked by $IP_3$, whether delivered from endogenous signalling pathways or uniformly to the cytosol, occur at immobile $IP_3R$ puncta close to the PM.

**STIM1 accumulates at junctions adjacent to immobile $IP_3Rs$.** SOCE is stimulated when loss of $Ca^{2+}$ from the ER causes STIM to translocate to ER-PM junctions, where it stimulates opening of

Orai channels[8]. Since most $IP_3R$-mediated $Ca^{2+}$ puffs occur at immobile $IP_3R$ puncta close to the PM, we asked whether immobile puncta reside near the ER-PM junctions where SOCE occurs. We used TIRFM and immunostaining to detect formation of native STIM1 puncta at ER-PM junctions after treatment with thapsigargin to deplete ER $Ca^{2+}$ stores. Assessed pixel-by-pixel, there was no significant co-localization of STIM1 and EGFP-$IP_3R1$ puncta (Fig. 7a, b), but immobile $IP_3R1$ puncta were often

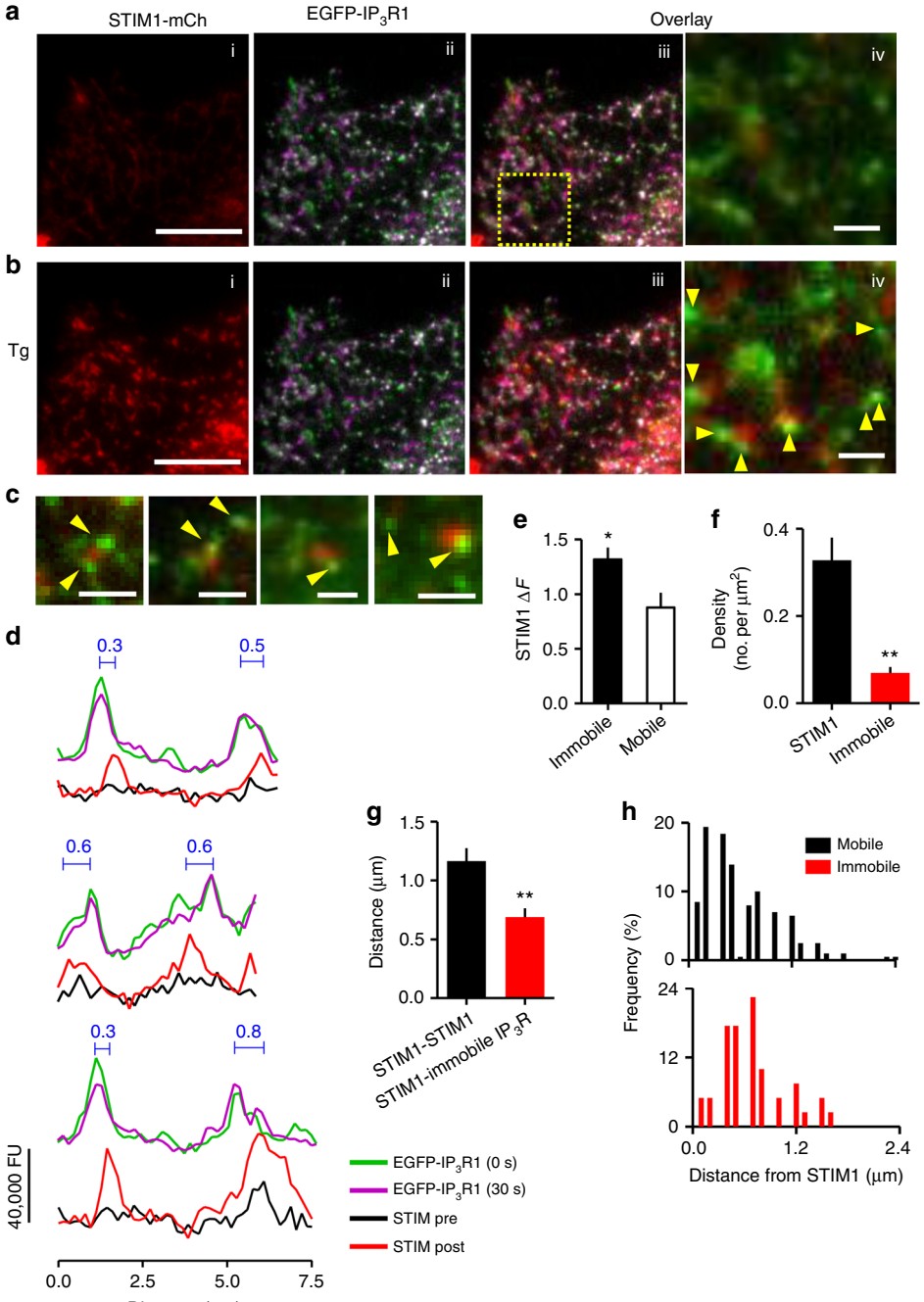

**Fig. 8** STIM1 translocates to ER-PM junctions adjacent to immobile IP₃R puncta. **a, b** TIRFM images show that thapsigargin (Tg, 1 μM, 5 min) causes formation of STIM1-mCherry puncta at ER-PM junctions (i). Immobile IP₃Rs (ii, white, identified from overlays of pseudocoloured EGFP-IP₃R distribution at 30-s intervals) abut STIM1 puncta (iii). Enlarged image of the boxed area shows all immobile EGFP-IP₃R1 (arrows) and STIM1-mCherry without pseudocolours for clarity (iv). Scale bars = 10 μm (i–iii), 2 μm (iv). **c** Similar overlay images from three additional Tg-treated cells, with all immobile IP₃R punta identified with arrows. Scale bars = 2 μm. **d** Fluorescence intensity profiles for EGFP-IP₃R1 measured at 30-s intervals (green and magenta) and STIM1-mCherry for transects drawn across Tg-treated cells. Results show that immobile IP₃Rs (white, where green and magenta coincide) and STIM1 after store depletion are juxtaposed, but not perfectly aligned. Distances (μm) between the peaks of the fluorescence intensity for STIM1 and immobile IP₃R are shown; 84 ± 9% (mean ± SD, $n = 3$ cells) of the centroids of immobile EGFP-IP₃R1 puncta were within twice their radius ($2r = 0.64$–$0.96$ μm) of a STIM1 punctum. **e** STIM1-mCherry fluorescence was measured before ($F_0$) and after Tg treatment ($F_{Tg}$) at ROI with twice the radius of underlying mobile or immobile EGFP-IP₃R1 puncta. Summary results show the change in mCherry fluorescence ($\Delta F = (F_{Tg}-F_0)/F_0$, mean ± SEM, $n = 3$ cells, 29–33 puncta). *$P < 0.05$, Student's $t$-test. **f, g** Density of STIM1-mCherry and EGFP-IP₃R1 puncta from randomly selected peripheral regions of cells (where puncta are most clearly separated) (**f**) and the separation between any STIM1 puncta identified within the randomly selected regions and the nearest STIM1 or immobile EGFP-IP₃R1 punctum (**g**). **h** Frequency distributions for the separations of STIM1 puncta from mobile and immobile EGFP-IP₃R1 puncta. Whereas 28% of mobile IP₃R puncta were within 300 nm of a STIM1 punctum, only 10% of immobile IP₃R puncta fell within this distance. Results (**f**–**h**) are from four cells. **$P < 0.01$, Student's $t$-test

closely apposed to STIM1 puncta (Fig. 7b–e). We confirmed[8] that in cells with empty $Ca^{2+}$ stores, mCherry-Orai1 and CFP-STIM1 were co-localized (Fig. 7f), suggesting that STIM1 puncta identify functional sites of SOCE[8].

In EGFP-IP$_3$R1 HeLa cells expressing STIM1-mCherry to allow real-time recording of EGFP-IP$_3$R1 mobility and STIM1 translocation, loss of $Ca^{2+}$ from the ER caused accumulation of STIM1 puncta at ER-PM junctions abutting immobile IP$_3$R puncta (Fig. 8a–d): $84 \pm 9\%$ (mean $\pm$ SD, $n = 3$) of immobile IP$_3$R puncta were juxtaposed to STIM1 puncta (see legend to Fig. 8d). The sites where STIM1 accumulated were selectively adjacent to immobile, rather than mobile, IP$_3$Rs (Fig. 8e). Although STIM1-mCherry puncta were much more abundant than immobile EGFP-IP$_3$R1 puncta (Fig. 8f), the average distance between STIM1 puncta was significantly larger than the distance between STIM1 and immobile EGFP-IP$_3$R1 puncta (Fig. 8g). The distribution of distances between STIM1 and IP$_3$Rs is consistent with mobile IP$_3$R puncta moving apparently freely around STIM1, while immobile IP$_3$Rs are typically $700 \pm 70$ nm away from STIM1 puncta and rarely within 300 nm (Fig. 8h). Similar results, namely accumulation of STIM1-mCherry puncta alongside immobile EGFP-IP$_3$R1 puncta, were obtained when histamine was used to evoke SOCE by stimulating $Ca^{2+}$ release from the ER through activation of IP$_3$Rs (Supplementary Fig. 18).

The close association of immobile IP$_3$Rs with the ER-PM junctions where STIM1 and Orai1 accumulate after store depletion (Figs 7, 8 and Supplementary Fig. 18) prompted us to ask whether IP$_3$Rs might contribute to assembly of the junctions. We expressed GFP-MAPPER, a STIM-derived non-perturbing marker of ER-PM junctions[46], in human embryonic kidney (HEK) cells with and without endogenous IP$_3$Rs[47]. The number and size of the junctions identified by GFP-MAPPER were indistinguishable in cells with and without IP$_3$Rs (Supplementary Fig. 19a–e). These results, which are consistent with evidence that IP$_3$Rs are not directly required for SOCE in vertebrate cells[48], demonstrate that IP$_3$Rs are not required for assembly of ER-PM junctions. Whereas GFP-MAPPER identified punctate ER-PM junctions (Supplementary Fig. 19b, c)[46], expression of the same protein with an mCherry tag (mCherry-MAPPER) caused massive expansion of the junctions. This was accompanied by displacement of immobile EGFP-IP$_3$R1 puncta to the margins of the enlarged junctions, apparently without preventing mobile IP$_3$Rs from passing through the regions enriched in junctions (Supplementary Movie 9 and Supplementary Fig. 19f). Similar results were obtained when mCherry-STIM1 was massively over-expressed (Supplementary Fig. 19g). These results suggest that the assembly of ER-PM junctions determines the location of immobile IP$_3$R puncta, rather than vice versa.

## Discussion

Using gene editing to tag endogenous IP$_3$R1 with EGFP, we provide measurements of the distribution of endogenous IP$_3$Rs while simultaneously recording the $Ca^{2+}$ signals they evoke. Our results establish that within ER membranes IP$_3$Rs form small clusters that typically include about eight tetrameric IP$_3$Rs (Fig. 1). The estimate is approximate because it requires several assumptions (Supplementary Fig. 5), but it confirms that clusters include only a few IP$_3$Rs. Our estimate is similar to the small numbers suggested by STORM images (Fig. 4), by single-particle tracking of over-expressed mEos2-IP$_3$R1[17] and from functional analyses of $Ca^{2+}$ puffs, which suggest there are about 2–9 functional IP$_3$Rs within most puff sites[45]. We suggest that these small stable clusters of IP$_3$Rs are the elementary structural units for IP$_3$-evoked $Ca^{2+}$ signalling. Within clusters, many IP$_3$Rs are too far apart for them to interact directly, suggesting that additional

scaffolds maintain the loose affiliation of IP$_3$Rs within clusters (Fig. 4). The scaffolds may be provided by lipid microdomains, cytosolic or ER proteins, or by interactions with other organelles such as lysosomes or mitochondria with which IP$_3$Rs form intimate contacts[49]. These scaffolds are likely to be important in determining whether IP$_3$Rs can sustain local propagation of $Ca^{2+}$ signals by $Ca^{2+}$-induced $Ca^{2+}$ release, and in segregating populations of mobile and immobile IP$_3$Rs.

Most IP$_3$R puncta are mobile within ER membranes ($M_f \sim$ 74%), but a smaller fraction is immobile over periods of many minutes (Figs 2, 3, 5). These results are consistent with FRAP analyses of over-expressed EGFP-IP$_3$R1[16] and EGFP-IP$_3$R3[15], the major subtypes in HeLa cells. However, our analyses unmasked a pool of IP$_3$Rs (~7%) that move directionally (Fig. 5), and demonstrated that kinesin-1 and dynein contribute to this movement (Supplementary Figs 11–13). A study using mEos2-IP$_3$R1 expressed in COS-7 cells suggested a similar $M_f$ to that of EGFP-IP$_3$R1 in HeLa cells (~70%), but detected no directional movement of IP$_3$Rs[17]. The difference may be due to the different cell types or to over-expressed mEos2-IP$_3$R1s masking movement of the small number of IP$_3$Rs moved by microtubule motors. In neurons, proteins may be transported along dendrites by microtubule motors in either ER-derived vesicles[42,50] or as a bolus of proteins within ER membranes[51–53]. The latter seems most likely for directed movement of EGFP-IP$_3$R1 in HeLa cells (Supplementary Fig. 14). This directed, motor-driven transport may allow relocation of IP$_3$Rs within cells that rely on IP$_3$R-mediated $Ca^{2+}$ release to control local events, such as migration[54] and growth of neurons[55].

Over many minutes there is minimal exchange of IP$_3$Rs between mobile and immobile puncta (Fig. 3). Since only immobile IP$_3$Rs initiate $Ca^{2+}$ signals (Fig. 6 and Supplementary Fig. 17), it will be interesting to resolve whether physiological stimuli regulate exchange between these pools and perhaps thereby influence IP$_3$ sensitivity.

By simultaneously measuring cytosolic $Ca^{2+}$ signals and the dynamics of native IP$_3$Rs, we show that almost all $Ca^{2+}$ puffs originate from immobile clusters of IP$_3$Rs adjacent to the PM (Fig. 6 and Supplementary Figs 15–17). This occurs whether IP$_3$ is delivered uniformly to the cytosol or immediately beneath the PM from endogenous signalling pathways. Since only about 30% of IP$_3$R puncta are immobile (Fig. 5g) and only a tiny fraction of them are close to the PM (Supplementary Fig. 6), our results establish that there is an additional level of IP$_3$R regulation that licenses a very small number of IP$_3$Rs to respond to IP$_3$. The larger number of IP$_3$Rs within some immobile clusters (Fig. 2f, g) may contribute to their propensity to respond, but it is probably not the only factor (see Results). An important issue is to resolve the mechanisms that provide near-PM immobile IP$_3$Rs with the competence to respond to IP$_3$.

The responsive immobile IP$_3$Rs are adjacent to, but not coincident with, the ER-PM junctions that become populated by STIM1 and Orai after depletion of ER $Ca^{2+}$ (Figs 7, 8 and Supplementary Figs 18, 19). A likely explanation is that the gap between the ER and PM across which STIM and Orai interact is probably too narrow (about 9–21 nm)[9,14,56] to accommodate the immobile IP$_3$Rs, which project about 15 nm from ER membranes[34]. This explanation would be consistent with enforced exaggeration of the ER-PM junctions causing immobile IP$_3$R puncta (presumably in ER facing the PM) to be excluded from the junctions, without preventing mobile IP$_3$Rs (populating both sides of the ER) from appearing to pass through them (Supplementary Movie 9 and Supplementary Fig. 19f, g). We speculate that responsive immobile IP$_3$R puncta reside in the ER facing the PM, alongside the ER-PM junctions where STIM1 and Orai interact, but they are excluded from the junction itself. We

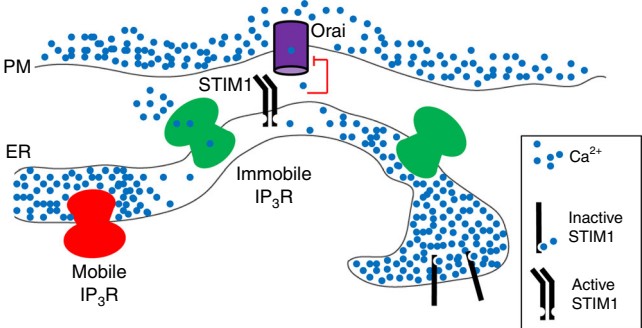

**Fig. 9** $Ca^{2+}$ signals evoked by SOCE occur alongside immobile IP$_3$Rs. Clustered IP$_3$Rs evoke $Ca^{2+}$ puffs when they bind IP$_3$ and then respond to $Ca^{2+}$ released by a neighbouring IP$_3$R. Only a small fraction of all IP$_3$Rs, namely immobile clusters close to the PM, are licensed to respond (green). The licensed IP$_3$Rs are adjacent to the junctions where STIM1 and Orai interact to mediate SOCE, which is itself locally regulated by $Ca^{2+}$ passing through Orai (red lines). We suggest that the juxtaposition of responsive IP$_3$Rs and the SOCE machinery allows local depletion of ER $Ca^{2+}$ stores to activate SOCE, while sparing it from inhibition by the large $Ca^{2+}$ fluxes through IP$_3$Rs

suggest that this juxtaposition of the signalling proteins that mediate $Ca^{2+}$ release and $Ca^{2+}$ entry has important implications (Fig. 9). First, PIP$_2$ in the PM recruits active STIM1 to ER-PM junctions. Hence, licensed IP$_3$Rs are adjacent to the substrate from which IP$_3$ is generated. Second, activation of STIM1 and SOCE requires substantial depletion of ER $Ca^{2+}$ stores. The licensed IP$_3$Rs are well placed to locally deplete the stores closest to the ER-PM junctions where SOCE occurs. Third, inhibition of SOCE by increased cytosolic $Ca^{2+}$ concentration provides local feedback regulation of Orai gating[7,57]. Hence, IP$_3$Rs, with their very large $Ca^{2+}$ conductance, might inhibit SOCE if they were within ER-PM junctions. We speculate that licensed IP$_3$Rs are optimally placed to locally deplete the ER, allowing it to activate SOCE effectively, but far enough away to avoid $Ca^{2+}$-mediated inactivation of SOCE and disruption of local feedback regulation (Fig. 9).

We have shown that native IP$_3$Rs are scaffolded into small clusters within ER membranes. Most IP$_3$R clusters are mobile, but it is the small number of immobile IP$_3$R clusters sitting alongside the ER-PM junctions where SOCE occurs that are licensed to respond to IP$_3$ and initiate $Ca^{2+}$ signals. We suggest that this arrangement of IP$_3$Rs may allow effective delivery of IP$_3$ from endogenous signalling pathways and local regulation of SOCE.

## Methods

**Materials.** BAPTA was from Molekula. Cal-590 AM, Cal-520 AM and Fluo-8 AM were from AAT Bioquest. Ciliobrevin D, EGTA-AM, human fibronectin and nocodazole were from Merck Millipore. A membrane-permeant form of caged-IP$_3$ (ci-IP$_3$/PM: D-2,3-O-isopropylidene-6-O-(2-nitro-4,5-dimethoxy)benzyl-myo-inositol 1,4,5-trisphosphate hexakis(propionoxymethyl) ester) was from SiChem. Mycalolide B was from Santa Cruz Biotechnology. Rapamycin was from Cell Guidance Systems. Histamine and carbachol were from Sigma-Aldrich. Bovine serum albumin (BSA) was from Europa Bio-Products.

Sources of plasmids encoding the following proteins were: mCherry-EB3 (Addgene #55038); mCherry-C1 (Clontech #632524); GFP-ER$_{memb}$ (GFP targeted to ER membranes via the ER-targeting sequence of yeast UBC6 protein)[36]; p50-mCherry (dominant-negative cytoplasmic dynein tagged with mCherry)[36]; RFP-KHC-CT (dominant-negative kinesin-1 tagged with RFP)[36]; pBa-KIF5C 559-tdTomato-FKBP (kinesin-1 tagged with tandem dimer Tomato and FKBP to allow dimerization with Lyn11-FRB-CFP via rapamycin) (Addgene #64211)[38]; GFP-TPC2[58]; mCherry-ER (Addgene #55041); Lyn11-FRB-CFP (PM-targeted FRB tagged with CFP to allow dimerization with FKBP via rapamycin) (Addgene #38003); LAMP1-mCherry[59]; LAMP1-AcGFP[59]; STIM1-mCherry[60]; CFP-STIM1 (Addgene #18858); mCherry-tubulin (Addgene #26768); GFP-MAPPER[46] and Orai1-CFP (Addgene #19757). The mCherry-MAPPER construct was generated by

amplifying mCherry from pmCherry-C1 (Clontech) by PCR using forward (CGCTAGCGCTACCGGTC) and reverse (AGACTAGTTGATCCGGACTTG TACAGCTCGTC) primers, digesting the resulting product with AgeI/SpeI and inserting the product into similarly digested GFP-MAPPER. For the mCherry-Orai1 construct, human Orai1 was amplified by PCR from hOrai1-pcDNA6[61] using forward (GGAGGGGGATCTATGCATCCGGAGCCCG) and reverse (AGCGGCCGCCACTGTGCTGGATATCTGCAGCTAGGCATAGTGGCT GCCG) primers, and mCherry was amplified from pmCherry-C1 (Clontech) using forward (CGTTTAAACTTAAGCTTGGTACCGAGCTCAAGCTTGCCACCATG GTGAGCAAGGGCGAG) and reverse (CGGGCTCCGGATGCATAGATCCC CCTCCCTTGTACAGCTCGTCCATGCC) primers. The resulting PCR products were introduced by Gibson assembly (Gibson Assembly Master Mix; New England Biolabs) into pcDNA3.1+ digested with BamHI/EcoRI.

Primers used for sequencing and amplification were from ThermoFisher: forward mCherry (mCherry PF: ATGGTGAGCAAGGGCGAG); reverse itpr1 (P5R: GGGGGCCTCACTACTCCTTTC); forward itpr1 (P2F: GGGTGACATG CGTACAGTTG); reverse itpr1 (P4R: GAACGTCACCGTTTTACTTGG); forward itpr3 (P13F: CTCCAAGCCTGAGCACTTTC); reverse itpr3 (P16R: CCCGATAGAGGCCTGGAC). The TALENs used are listed in Supplementary Fig. 1b. Antibodies (for Western blotting, WB; immunoprecipitation, IP) were from the following suppliers: β-actin (WB 1:1,000; Abcam, ab6276), dynein, mouse monoclonal (IP and WB 1:1,000, Abcam, ab23905); GFP (WB 1:1,000, ChromoTek, #3H9); GFP Tag-Alexa Fluor-647 (STORM 1:400, ThermoFisher, #31852); GFP-Trap, anti-GFP V$_H$H coupled to magnetic beads (IP 100 μl per 200 μl lysate, ChromoTek, #gtm20); IP$_3$R1 (rabbit, C-terminal peptide, WB 1:1,000)[62]; IP$_3$R2 (rabbit, C-terminal peptide GFLGSNTPHENHHMPPH, WB 1:1,000, Pocono Rabbit Farm and Laboratory); IP$_3$R3 (WB 1:1,000, BD Transduction Laboratories, #610313); kinesin-1 (WB 1:1,000, IP 1:400, Abcam, ab62104); mCherry (WB 1:1,000, Abcam, ab167453); STIM1 (New England Biolabs, D88E10); RFP-Trap, anti-RFP V$_H$H coupled to magnetic beads (IP 100 μl per 200 μl lysate, ChromoTek, #rtm10); donkey anti-rabbit IgG-HRP (WB 1:5,000, Santa Cruz, SC-2313); donkey anti-mouse IgG-HRP (WB 1:2,000, Santa Cruz, SC-2314); goat anti-rat IgG-HRP (WB 1:5,000, Santa Cruz, SC-2020); goat anti-rabbit Alexa Fluor-594 (ThermoFisher, A11012).

Additional sources of materials are provided within the relevant methods.

**Cell culture and transient transfection.** HEK293 and HeLa cells (both from ATCC) were cultured in Dulbecco's modified Eagle's medium/F-12 with Gluta-MAX (ThermoFisher) supplemented with fetal bovine serum (FBS, 10%, Sigma). The cells were maintained at 37 °C in humidified air with 5% $CO_2$, and passaged every 3–4 days using Gibco TrypLE Express (ThermoFisher). For imaging, cells were grown on 35-mm glass-bottomed dishes (#P35G-1.0-14-C, MatTek) coated with human fibronectin (10 μg ml$^{-1}$). HeLa cells and EGFP-IP$_3$R1 HeLa cells were transfected, according to the manufacturer's instructions, with TransIT-LT1 (GeneFlow) and ViaFect (Promega) reagents, respectively (1 μg DNA per 2.5 μl reagent). Short tandem repeat profiling was used to authenticate HeLa cells (Eurofins, Germany) and HEK293 cells (Public Health England). Screening confirmed that all cells were free of mycoplasma infection.

**Gene editing of IP$_3$R1 in HeLa cells.** The methods are summarized in Supplementary Fig. 1. TALENs were designed to target the first coding exon (exon 3) of the human itpr1 gene at the 3p26.1 locus. A double-stranded donor DNA was synthesized (ThermoFisher) with homology regions for recombination with the itpr1 gene (~300 bp either side of the TALEN cleavage site, with SalI and XhoI restriction sites at the 5′ and 3′ ends, respectively) flanking a sequence encoding EGFP mutated (A206K) to ensure the expressed protein was monomeric. The donor DNA was cloned into the pENTR1A-dual selection vector (ThermoFisher), excised from it with SalI and XhoI, and purified to provide the donor DNA used for transfection.

HeLa cells (75-cm$^2$ flask, ~60% confluence) were cotransfected with double-stranded donor DNA (9 μg) and the TALENs (6 μg of each) using TransIT-LT1. Cells were harvested after 48–72 h using trypsin, washed in phosphate-buffered saline (PBS: 1.06 mM KH$_2$PO$_4$, 155 mM NaCl, 3 mM Na$_2$HPO$_4$, pH 7.3) containing FBS (1%) and re-suspended (~$10^7$ cells ml$^{-1}$) in sorting buffer (PBS with 1 mM EDTA, 25 mM HEPES, 1% FBS, pH 7.0). Cells were sorted by fluorescence-activated cell sorting (FACS, excitation 488 nm, emission 525 nm) using a modular flow multilaser sorter flow cytometer (DakoCytomation, Beckmann Coulter). Cells with the most intense EGFP fluorescence (top ~1%) were collected into growth medium containing penicillin (100 μg ml$^{-1}$), streptomycin (100 μg ml$^{-1}$), amphotericin B (0.25 μg ml$^{-1}$, all from ThermoFisher) and FBS (20%). Polyclonal cells were cultured in this medium for two passages, and then in normal growth medium without antibiotics. Two further rounds of FACS were used to enrich the EGFP-IP$_3$R1-expressing cells, which were then sorted as single cells into 96-well plates and cultured in the antibiotic-containing medium to select monoclonal cell lines (Supplementary Fig. 1c). Five monoclonal EGFP-IP$_3$R1 HeLa cell lines were propagated, one of which was used for the work reported here.

The sequences of the edited itpr1 locus of the five monoclonal EGFP-IP$_3$R1 HeLa cell lines was determined by isolating genomic DNA using a Quick-gDNA MiniPrep kit (Zymo Research). The itpr1 locus of the genomic DNA was amplified

by PCR using primers P2F and P4R, and the sequences were determined using the same primers. Sequencing confirmed the appropriate attachment of the EGFP sequence to IP$_3$R1 (Supplementary Fig. 2a).

Karyotype analysis of G-banded metaphase spreads of EGFP-IP$_3$R1 HeLa cells was performed by Cell Guidance Systems (Cambridge, UK) (Supplementary Fig. 2b).

**Measurements of [Ca$^{2+}$]$_c$ in cell populations.** EGFP-IP$_3$R1 HeLa cells were plated in clear-bottomed 96-well plates (Greiner Bio-One) coated with fibronectin (10 µg ml$^{-1}$). Cells were transfected with siRNA directed against GFP (50 nM, #AM4626, ThermoFisher) or a nonsilencing control siRNA (50 nM, #AM4611, ThermoFisher) using siPORT NeoFX transfection reagent (ThermoFisher, 220 ng siRNA per µl reagent). After 72 h, cells were washed with HEPES-buffered saline (HBS: 135 mM NaCl, 5.9 mM KCl, 1.2 mM MgCl$_2$, 1.5 mM CaCl$_2$, 11.6 mM HEPES, 11.5 mM glucose; pH 7.3), loaded with Fluo-8 by incubation in HBS with Fluo-8 AM (2 µM, 60 min, 20 °C), washed and incubated in HBS (60 min, 20 °C) before experiments. A FlexStation 3 microplate reader (Molecular Devices) was used to measure Fluo-8 fluorescence at 20 °C. After addition of BAPTA in Ca$^{2+}$-free HBS (final BAPTA and CaCl$_2$ concentrations = 2.5 and 0.75 mM, respectively; free [Ca$^{2+}$]<40 nM), histamine (100 µM) was added to stimulate IP$_3$ formation through endogenous H$_1$ histamine receptors. Fluorescence was collected (using SoftMax Pro, Molecular Devices) and calibrated to cytosolic free Ca$^{2+}$ concentration ([Ca$^{2+}$]$_c$) as described[63].

**Functional expression of EGFP-IP$_3$R1 in HEK cells.** An EGFP-IP$_3$R1 construct that replicates the linker produced by gene editing of endogenous IP$_3$R1 in HeLa cells (Supplementary Fig. 2a) was transiently expressed in HEK cells in which all endogenous IP$_3$R genes had been disrupted (Kerafast, Boston, MA, USA)[47]. The plasmid, EGFP-hIP$_3$R1-pcDNA3.1(+), was constructed as follows. The open reading frame of human IP$_3$R1 (NCBI reference sequence NM_002222.5), with the sequence (GGATCCGGTACCGAGGAGATCTGCCGCCGCGATCGCC) immediately upstream of the start codon and the sequence (ACGCGTACGCGGCCGCTCGAGGA) immediately downstream of the stop codon, was synthesized and subcloned into the SmaI/SalI sites of pUC57 (Bio Basic, Markham, Ontario, Canada). The sequence was then transferred into pcDNA3.1(+) using KpnI/NotI. The EGFP donor-pENTR1A plasmid used for gene editing was amplified by PCR using forward (CTGGCTAGCGTTTAAACTTAAGCTTGGTACATGGTG AGCAAGGGCGAG) and reverse (AGGAAGCTAGACATTTTGTCAGACA- TACCAGAACCACCACCAGAAC) primers. The product was inserted using Gibson Assembly (NEBuilder HiFi DNA Assembly Master Mix, New England BioLabs) into hIP$_3$R1-pcDNA3.1(+) digested with KpnI/SfaAI, to give EGFP- hIP$_3$R1-pcDNA3.1(+). Ca$^{2+}$ signals in single cells were measured using the genetically encoded Ca$^{2+}$ sensor, RGECO1[64].

**In-gel fluorescence and WB.** Cells were harvested using either Cell Dissociation Buffer (ThermoFisher) or, for siRNA experiments, by scraping cells into lysis medium (150 mM NaCl, 0.5 mM EDTA, 1% Triton X-100, 10 mM Tris/HCl (pH 7.5), Pierce protease inhibitor mini-tablet with EDTA, 1 tablet per 10 ml). Cells were lysed by incubation in the same medium at 4 °C for 1 h followed by sonication (Transonic ultrasonic bath, 3 × 10 s), and centrifugation (20,000 × g, 30 min) to produce the supernatant used for subsequent analyses.

For in-gel fluorescence, proteins were separated using Novex 4–12% Tris- glycine mini protein gels (ThermoFisher). EGFP fluorescence was detected using a PXi chemiluminescence gel-imaging system (Syngene) equipped with a light- emitting diode (465 nm) for excitation and a filter for emission (525 nm).

For WB, proteins were separated using NuPAGE 3–8% Tris-acetate protein gels, and transferred onto an iBlot PVDF membrane using an iBlot gel-transfer device (ThermoFisher). The membranes were blocked with BSA (5%) in Tris- buffered saline (TBS: 137 mM NaCl, 20 mM Tris, pH 7.6) containing 0.1% Tween- 20, incubated with the primary antibody (~16 h at 4 °C), washed (3 × 5 min), and incubated with HRP-conjugated secondary antibody (1 h at 20 °C) (see Materials for the antibodies used). After further washes (3 × 5 min), HRP was detected using ECL Prime Western blotting detection reagents (GE Healthcare Life Sciences) and a PXi chemiluminescence detection system. Scans of the most important WB and gels are shown without cropping in Supplementary Fig. 20.

**Immunoprecipitation.** GFP-Trap magnetic beads (ChromoTek) were used to immunoprecipitate EGFP-IP$_3$R1 according to the manufacturer's protocol. Briefly, supernatants (200 µl) from cell lysates prepared as for WB were incubated with a slurry of GFP-Trap magnetic beads (60 µl). After 2 h at 4 °C, immunocomplexes were separated magnetically, washed (3 × 500 µl wash buffer: 150 mM NaCl, 0.5 mM EDTA, 10 mM Tris, pH 7.5) and dissociated from the beads by boiling (10 min, 95 °C) in 2× LDS sample buffer (ThermoFisher). Samples were then separated by sodium dodecyl sulfate-polyacrylamide gel electrophoresis and sub- jected to WB.

For immunoprecipitation of kinesin-1 and dynein, protein G magnetic Dynabeads (~1.5 mg in 200 µl PBS, ThermoFisher) were conjugated to antibodies (~10 µg) against kinesin-1 or dynein according to the manufacturer's instructions. Lysates were pre-cleared by incubation with either magnetic particles

(ChromoTek) or protein G beads (ThermoFisher) for 1 h at 4 °C. The conjugated beads (1.5 mg) were then incubated with 200 µl of the pre-cleared lysate supernatant for ~16 h at 4 °C. After washing (PBS, 3 × 200 µl), samples were resuspended in elution buffer (20 µl of 50 mM glycine (pH 2.8) and 10 µl NuPage LDS sample buffer), proteins were eluted (70 °C, 10 min) and the beads were removed magnetically. Samples were then separated by sodium dodecyl sulfate- polyacrylamide gel electrophoresis and subjected to WB.

**Fluorescence microscopy.** All fluorescence microscopy used an inverted Olympus IX83 microscope equipped with a 100× oil-immersion TIRF objective (numerical aperture, NA 1.49), a multi-line laser bank (395 nm (for FRAP), 425 nm, 488 nm, 561 nm and 647 nm) and an iLas2 targeted laser illumination system (Cairn, Faversham, Kent, UK). Excitation light was transmitted through either a quad dichroic beam splitter (TRF89902-QUAD) or a dichroic mirror (for 425 nm; ZT442rdc-UF2) (Chroma). Emitted light was passed through appropriate filters (Cairn Optospin; peak/bandwidth: 480/40, 525/50, 630/75 and 700/75 nm) and detected with an iXon Ultra 897 electron multiplied charge-coupled device (EMCCD) camera (512 × 512 pixels, Andor).

Spinning disc confocal microscopy used a spinning disk with a 70-µm pinhole (X-Light, Crest Optics). For TIRFM, the penetration depth was 90–140 nm. The iLas2 illumination system was used for TIRFM, wide-field imaging and FRAP. Bright-field images were acquired using a Cairn MonoLED illuminator.

Before analysis, all fluorescence images were corrected for background by subtraction of the fluorescence detected from a region outside the cell. Image capture and processing used MetaMorph Microscopy Automation and Image Analysis Software (Molecular Devices) and Fiji[65]. Co-localization analysis was carried out using the JACoP ImageJ plugin[66].

**Immunocytochemistry.** Cells grown on fibronectin-coated MatTek dishes were washed five times in PBS, fixed with paraformaldehyde (4%) in PBS (30 min, 20 °C), washed twice in PBS with gentle agitation, and then permeabilized by incubation (5 min, 20 °C) in PBS containing 0.25% Triton X-100 with gentle agitation. After washing twice with PBS, cells were incubated (1 h, 20 °C) in blocking buffer (5% BSA in PBS), washed twice, and then incubated with 1:400 anti-GFP antibody-Alexa Fluor-647 conjugate (1:400) in PBS for 1 h at 20 °C with gentle agitation. For STIM1 immunostaining, cells were incubated with anti-STIM1 antibody (1:800) for 16 h at 4 °C, washed three times with PBS, and then incubated with goat anti-rabbit Alexa Fluor-594 secondary antibody (1:1,000) for 1 h at 20 °C. The cells were then washed and imaged.

**Measurements of local Ca$^{2+}$ signals in single cells.** EGFP-IP$_3$R1 HeLa cells grown on fibronectin-coated MatTek dishes were washed twice with HBS and incubated (37 °C, 5% CO$_2$) in HBS containing Cal-590-AM (2 µM) alone or with ci-IP$_3$/PM (1 µM). After 1 h, the cells were washed twice with HBS, and then incubated for 30 min at 20 °C in HBS containing EGTA-AM (5 µM). EGTA is a slow Ca$^{2+}$ buffer used to limit regenerative propagation of Ca$^{2+}$ signals without perturbing local Ca$^{2+}$ release events[4]. The cells were then washed twice with HBS, and incubated for 30 min in HBS at 20 °C to allow de-esterification of the esterified Cal-590, EGTA and caged-IP$_3$. The medium was replaced with fresh HBS before imaging. All incubations prior to imaging were in the dark.

Cal-590 is the only practicable Ca$^{2+}$ indicator compatible with imaging of EGFP, but it has lower affinity for Ca$^{2+}$ ($K_D = 561$ nM) and it is less bright than Fluo-8 ($K_D = 389$ nM). These limitations and the need to minimize photobleaching constrain opportunities to resolve the temporal dynamics of Ca$^{2+}$ puffs, but they do not prevent their accurate spatial localization. In experiments designed to capture the spatial distribution of Ca$^{2+}$ signals and EGFP-IP$_3$R1, it was impracticable to collect fully interleaved images of EGFP and Cal-590 fluorescence at sufficient temporal resolution to identify all Ca$^{2+}$ puffs. We therefore adopted two approaches. In the first, we collected interleaved TIRFM images of EGFP-IP$_3$R1 and Cal-590 fluorescence at relatively low temporal resolution (700 ms between successive measurements at the same wavelength). This allowed the distribution of both mobile and immobile IP$_3$Rs to be compared with almost coincident Ca$^{2+}$ signals (Supplementary Fig. 17). In other experiments (Fig. 6 and Supplementary Fig. 15), we identified EGFP-IP$_3$R1 (~3 s, 488-nm excitation, 525/50-nm emission, 5 frames s$^{-1}$, fps) immediately before and after fast imaging of Ca$^{2+}$ signals. From the TIRF images of EGFP-IP$_3$R1, we then identified immobile puncta (Supplementary Fig. 7). Cal-590 fluorescence was recorded by TIRFM for 2–3 s (561-nm excitation, 20 fps, with illumination for 25 ms of every 50-ms capture interval to minimize bleaching) before collecting further images (~30 s) after addition of histamine (10 µM) or photolysis of ci-IP$_3$ (using SPECTRA X-light engine, Lumencor, 395/20 excitation, exposure time 50–75 ms). Spinning disc confocal microscopy with Cal-520 was used to image Ca$^{2+}$ puffs across confocal sections of wild-type HeLa cells. Images were collected using MetaMorph, corrected for background fluorescence and Ca$^{2+}$ puffs were detected and analysed using xySpark, a Fiji plugin[67].

**Fluorescence recovery after photobleaching.** For wide-field FRAP, time-lapse images (2 fps) of EGFP fluorescence were acquired using a 488-nm laser. A region of interest (ROI, radius = 1.84 µm) was rapidly (225 ms) photobleached by raster

scanning using the iLas2 laser illumination system (395 nm). A FRAP macro (https://www.med.unc.edu/microscopy/resources/imagej-plugins-and-macros/frap-calculator-macro)[68] was used to calculate the mobile fraction ($M_f$) and half-time ($t_{1/2}$) for recovery (ImageJ) from images corrected for background fluorescence. The analysis corrects for loss of fluorescence from both focal bleach of the ROI and photobleaching during image acquisition by expressing fluorescence intensity within the ROI as a fraction of the fluorescence intensity from the entire cell at each time point. The corrected ROI fluorescence was then normalized to its pre-bleach value ($F_0$). The diffusion coefficient ($D$) was calculated from: $D = \frac{r^2 \gamma}{4 t_{1/2}}$, where $r$ is the radius of the bleached area (1.84 μm) and $\gamma$ is a correction factor for bleaching (~1 for a circular beam). The mobile fraction ($M_f$) was calculated from: $M_f = F_{300}/F_0$, where $F_{300}$ is the normalized fluorescence recorded after 300 s, when fluorescence recovery had reached a plateau.

FRAP was also used to determine whether EGFP-IP$_3$R1s exchange between mobile and immobile puncta. For these FRAP analyses, we define an immobile punctum as one for which there is no detectable movement in frames separated by 30 s. Since STORM analyses required slightly shorter (10 s) intervals between the images used to define immobile puncta, we note that the same immobile puncta were identified when the interval between images was varied between 5 and 60 s (Supplementary Fig. 7). Immobile puncta were identified by overlaying TIRFM images of EGFP fluorescence collected 30-s apart; the images were then pseudocoloured (green for $t = 0$, magenta for $t = 30$ s), such that in overlaid images immobile puncta appeared white. This means of defining immobile puncta is important because over the period (60 min) required to monitor fluorescence recovery, there is some re-arrangement of the ER. Hence, even immobile puncta may be in slightly different positions after 60 min, making it impractical to determine their fluorescence recovery by recording from pre-defined regions. For each time during the recovery period, therefore, immobile puncta were uniquely identified from the appropriate pseudocolour images.

For FRAP analysis of the exchange of EGFP-IP$_3$R1 between mobile and immobile puncta, a rectangular strip of the cell (typically ~30 μm × 7 μm, and comprising ~10% of the cell area) was photobleached by raster scanning (1.25 s, wide-field using the iLas2 laser 395-nm illumination system). By bleaching a large area, it was possible to monitor recovery of both mobile puncta and many immobile puncta within it. Fluorescence images were then collected using TIRFM (488 nm) at 30-s intervals. After correction for background fluorescence, ROIs corresponding to immobile puncta within the bleached area were identified and pooled for subsequent analysis. The remaining bleached area (with ROI corresponding to immobile puncta removed) was analysed separately. Recovery from photobleaching was calculated from $F/F_0$.

**Stochastic optical reconstruction microscopy**. Prior to fixation for STORM, TIRFM of EGFP-IP$_3$R1 HeLa cells in HBS was used to identify mobile and immobile puncta, using the methods described in the FRAP section, but with the interval between successive images reduced to 10 s (Supplementary Fig. 7). Cells were then immediately fixed *in situ* by replacing the HBS with PBS containing paraformaldehyde (4%, 30 min). The cells were then permeabilized and stained with Alexa Fluor-647-conjugated anti-GFP antibody as described in the Immunocytochemistry section. These methods preserved the original distribution of EGFP-IP$_3$R1 puncta (Supplementary Fig. 9). The medium used for STORM[69] included an oxygen-scavenging system to reduce photobleaching, 2-mercaptoethanol to improve photoswitching and cyclooctatetraene to improve photon yield; it comprised Tris-HCl (50 mM), NaCl (10 mM) glucose (100 g l$^{-1}$), pH 8.0), catalase (34 μg ml$^{-1}$), glucose oxidase (0.56 mg ml$^{-1}$), 2-mercaptoethanol (1%) and cyclooctatetraene (1%). For STORM, the originally characterized cell was re-identified, and Alexa Fluor-647 fluorescence was bleached using the full laser power (647 nm) in semi-TIRF mode until individual fluorophores began to blink. Images (30,000 at 50 fps) were then acquired by TIRFM using a TIRF objective (×100, NA = 1.49 with an intermediate magnification of ×1.6) during continuous illumination with a 647-nm laser and captured (256 × 256 pixels) with an EMCCD camera (Andor iXon897 Ultra). Image collection and fitting used WaveTracer (MetaMorph), which sets a photon-detection threshold to identify genuine blinking events and a wavelet segmentation algorithm to fit a centroid to each blinking event within each 10 nm × 10 nm pixel[70]. Point locations of individually blinking fluorophores in each of the 30,000 frames were collated to form a super-resolution image. TetraSpeck microspheres (0.1 μm diameter, Thermo-Fisher) were used to verify that there was no significant drift during image acquisition.

Analysis of the distribution of fluorophores within puncta used the program SR-Tesseler[71], which effectively encloses every blinking event (= localization) within a field (Voronoi tessellation), the boundaries of which are drawn to be equidistant between each localization and its neighbours. Hence, the areas of these fields are smaller (and so the density of localizations is larger) in regions where many events are detected. From the STORM images we manually re-identified immobile puncta, and within the ROIs corresponding to each mobile and immobile punctum, we set a density threshold (1.2× the average localization density within the ROI) to delineate the area (A1) within which fluorophores were concentrated. A second round of thresholding (1.2× the average density of A1) was used to define distributions of fluorophores within A1.

**Automated detection of fluorescent puncta**. The Fiji TrackMate plugin[72] was used to identify fluorescent puncta in background-corrected TIRFM images of EGFP-IP$_3$R1 HeLa cells, using a 'difference of Gaussians' filter after applying a consistent threshold. The algorithm detected no puncta in wild-type HeLa cells, and identified all discernible puncta in EGFP-IP$_3$R1 HeLa cells without false-positives (Supplementary Fig. 4). The automated detection of puncta attributed 34 ± 4% ($n = 5$ cells, 2,171 puncta) of total background-corrected fluorescence to puncta.

**Single-particle tracking**. The Fiji TrackMate plugin[72] was used to track single particles in background-corrected time-lapse TIRFM images (10 fps for 10 s). Using a parabolic interpolation scheme, the sub-pixel localization of each detected diffraction-limited spot was obtained. Track segments were generated by frame-to-frame linking of particles and the segments were linked by gap-closing. A linear assignment problem tracking algorithm[73] was used to track the particles. Our user-defined parameters were: maximum distance for frame-to-frame linking (400 nm, 5 pixels; selected after preliminary analysis of manual tracks to avoid incorrect linking of particles); gap-closing (600 nm; maximal allowable distance travelled between frames 1 and 3, when the particle is not detected in frame 2); split and merge events (300 nm; the maximal allowable distance from a particle splitting or merging and its subsequent detection as a new particle). We considered particles that moved >40 nm between successive frames (100 ms) to be mobile because in fixed cells, we detected a mean displacement of particles between frames of 40 nm.

TraJClassifier (Fiji plugin), which has been extensively validated using computational models and experimental data[74], was used to classify the mobility of single-particle trajectories (Fig. 5g and Supplementary Fig. 10). This algorithm uses a random forest ensemble learning approach to classify trajectories into diffusive, sub-diffusive, confined and directed trajectories. When a particle changes its behaviour during the recording, TraJClassifier splits its trajectory into sub-trajectories. Only 133 of the 4,492 trajectories analysed were split into sub-trajectories. Hence, only 3% of puncta changed their form of motion during the >3 s recording. A minimum trajectory of 30 frames (3 s) was used for analysis to minimize the problems associated with analysis of very short trajectories[35]. We note that this automated algorithm does not separate immobile puncta from those (sub-diffusive) that move more slowly than diffusing puncta.

**Single-step photobleaching analysis**. Time-lapse TIRFM images were captured while alternating (50 ms each) between wide-field (to accelerate bleaching) and TIRF illumination. The TIRFM images were then used for analysis of single-step photobleaching of EGFP-IP$_3$R1. We attempted automated stepwise photobleaching analysis using the Progressive Idealization and Filtering (PIF) algorithm[75], but this systematically underestimated the number of bleaching events in the brightest puncta (not shown). We therefore performed manual analysis of photobleaching on randomly selected puncta using Fiji Time Series Analyser, v.2.0. The number of bleaching steps was computed from the initial fluorescence intensity of each punctum and the amplitude of the final bleaching step (Fig. 1e). We confirmed that neither fixation nor selection of particles in which the final bleaching step could be resolved affected the distribution of fluorescence intensities of the puncta (Supplementary Fig. 5).

**Analysis of IP$_3$Rs at ER-PM junctions**. EGFP-IP$_3$R1 HeLa cells transfected with plasmid encoding STIM1-mCherry were used 24 h after transfection. TIRFM images (EGFP, 488 nm; mCherry, 561 nm) were acquired (6 frames min$^{-1}$, 15 min) before and after addition of thapsigargin (1 μM) in HBS. After correction for background fluorescence (MetaMorph), immobile IP$_3$R puncta were identified (see FRAP section). The intensity of STIM1-mCherry was calculated (Fiji Time Series Analyser, v.2.0)[76] before and after thapsigargin treatment for circular ROIs centred on mobile or immobile IP$_3$R puncta and with twice the radius ($2r$) of the IP$_3$R puncta ($r = 0.32-0.48$ μm). For visualizing histamine-evoked STIM1 puncta, EGFP-IP$_3$R1 HeLa cells were transfected with plasmid encoding CFP-STIM1, and TIRFM images (CFP, 425 nm; EGFP, 488 nm) were acquired 24 h after transfection. We confirmed, using four-colour beads (TetraSpeck microspheres, 0.1 μm), that chromatic aberration did not contribute to the non-overlapping localization of EGFP-IP$_3$R1 and STIM1-mCherry, immunostained STIM1 or CFP-STIM1. Numbers of STIM1 puncta were calculated using Fiji TrackMate.

**Statistics**. Most results are presented as mean ± SEM from $n$ independent analyses. Statistical comparisons used paired or unpaired Student's $t$-tests, or analysis of variance with the Bonferroni correction used for multiple comparisons. Significance levels are shown as *$P < 0.05$, **$P < 0.01$, ***$P < 0.001$ and ****$P < 0.0001$.

**Data availability**. The authors declare that all data supporting the findings reported are presented within the article and associated Supplementary Information files. The data and materials are available on request from the corresponding authors.

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

## Acknowledgements

This work was supported by the Biotechnology and Biological Sciences Research Council, UK (L0000075) and the Wellcome Trust (101844). N.B.T. was supported by The Cambridge Commonwealth, European and International Trust. We thank V. Allen (Manchester) for plasmids for KHC-CT, p50-mCherry and GFP-ER, J. Liou (UT Southwestern, Dallas) for GFP-MAPPER plasmid, D. Beech (Leeds) for human Orai1 plasmid, R. Blunck (Montréal) for PIF software, F. Levet and J.-B. Sibarita (Bordeaux) for SR-Tesseler software, T. Wagner (Dortmund) for the TraJClassifier plugin and helpful discussions. We are grateful to M. Falke (Berlin) for advice.

## Author contributions

N.B.T. conducted most experiments. A.P.C. (TALEN methods), S.C.T. (microscopy) and D.L.P. (optical probes) contributed to experiments. N.B.T., D.L.P. and C.W.T. designed and analysed experiments, and wrote the paper. All authors reviewed the paper.
