## [Peer Review File · Nature Communications]

Reviewers' comments:

Reviewer #1 (Remarks to the Author):

In this study by Taylor and colleagues, gene editing was used to label endogenous IP3Rs and describe their organization, mobility, and function in HeLa cells. The authors present evidence that IP3Rs are clustered, move by diffusion and by microtubule-based motors, and that the only receptors that respond to IP3 are an immobile fraction located mostly next to ER-plasma membrane junctions. This last result is probably the most novel and implies that IP3Rs may cause local ER Ca²⁺ depletion that controls nearby store-operated Ca²⁺ channels.

The generation and validation of the GFP-IP3R cells was carefully and thoroughly done. Using this approach, the authors confirm several results previously observed for heterologously expressed IP3Rs, such as receptor mobility, clustering and preferred activity of immobile receptors. The two main new findings of the paper are that IP3Rs move in a directed fashion, with contributions from kinesin and dynein, and that immobile receptors located near STIM1 and ER-PM junctions are “licensed” to respond to IP3. A number of automated image analysis algorithms are used throughout the paper, but a major problem is that these tools are not validated for the system under study, which makes it difficult or impossible to judge the results. In many places, the conclusions are overstated based on the data that are shown, while in others too much space is devoted to data and analysis that do not add significantly to the paper. Most importantly, I think more data are needed to support the conclusion that IP3 and PI-linked receptors selectively activate immobile IP3Rs adjacent to STIM1 and ER-PM junctions.

Major comments

1. Number of IP3Rs in clusters.

Fig. 1 does show that IP3Rs are associated with the ER, that they occur in clusters of varying intensities, and that the 3 isoforms combine. Although these things are mostly already known from other studies, these data do validate the endogenous labeling. However, I did not find the quantitative analysis to be informative:

a. I don't see the point of the intensity quantitation in 1c and Supp Fig. 4a. The intensity distribution is a product of several factors, including number of receptors, number of GFPs per receptor, and position in the TIRF field. Given these complications (which the authors acknowledge), the presentation of these numerous histograms seems is descriptive without any

clear interpretation. In addition, later in the paper it becomes clear that the brightest puncta highlighted here do not serve any particular function. So what is the point of describing them in such detail?

b. The number of tetrameric IP3Rs per punctum is questionable. The analysis of step photobleaching grossly underestimates the number of steps. It is well known that because of the flickery nature of GFP fluorescence, it becomes increasingly difficult to resolve steps in particles having more than 3-4 GFPs. Thus, the mean and the distribution shown in Fig. 1h, i are not meaningful. In fact, when the peak fluorescence is divided by the mean amplitude of a single photobleaching event, the mean number is almost twice what the step counting indicates (Supp. Fig. 4). Representative automated fits to traces with 5-22 steps should be shown, as well as the percentage of puncta that could not be quantitated in this way, considering that the reliability of such fits is questionable.

c. The automatic particle detection method is critical in this paper (for IP3R clusters and STIM1 puncta) and must be validated by showing examples of what it does (and does not) classify as a punctum, and that it is detecting GFP and not punctate background fluorescence. This will greatly affect interpretations throughout the paper. Without such validation, the reader has to have blind faith that the algorithm is doing the right thing, which is not a given when signals are small and noisy.

d. In Fig. 2, step photobleaching data should be shown for bright vs dim puncta to support Fig. 2d, e. It is particularly important to state the percentage of puncta that could not be analyzed because of a lack of clearly defined steps. Because such puncta are more likely to have greater number of receptors, the method inherently underestimates the number. Because of these uncertainties, I do not think the GFP tagging method provides better quantitation of IP3R numbers than the Ca²⁺ puff method used by Parker's group (ref. 30).

2. The organization of IP3R in puncta imply a corral for IP3Rs.

In Fig. 4, the STORM analysis does not clearly delineate single IP3Rs. Please show a clear example of two IP3Rs separated by up to ~200 nm as stated in the text. It is possible that an immobile punctum such as presented in 4d is actually composed of multiple independent clusters in which the IP3R interact directly with each other, rather than being "corralled" by an unknown scaffold. Mobile puncta would be the best way to show movement of a collection of widely spaced IP3Rs, but there is only one example that is convincing (bottom panel in 4e). Perhaps more examples like that can be presented, rather than ones that show only a very small punctum with one receptor.

3. Analysis of IP3R mobility.

a. In Fig. 5, the classification of particle trajectories into diffusive, immobile, or directional

classes is problematic. It is well known that the stochastic nature of diffusion can create a bewildering variety of MSDs over short time periods, which can appear to be supra or sublinear (e.g., see Saxton, *Biophys J* 64:6, 1766-80). The method used for automatic sorting into these categories needs to be validated, e.g. by analyzing computational models of particles with known modes of mobility. How would particles alternating between diffusion and active transport be treated by the algorithm? In addition, more information can be recovered from the data. The authors should show the average MSD vs time for the population of particles characterized as diffusive; what D does it predict? For those that are directed, the MSD vs time should allow an estimate of the speed of the motor, and is this roughly consistent with kinesin and/or dynein turnover rates?

b. The evidence for motors in IP3R transport is good, but the analysis in Supp. Fig. 6 is somewhat confusing. I do not know why the number of directional trajectories should be reduced if one type of motor is inhibited; rather, I would think the direction or velocity would be affected. Rather than plotting the number of directional trajectories, which is subject to the accuracy of the TrackMate algorithm, why not simply plot the MSD vs time for the various motor treatments? This would show the contribution of motors in a more readily comprehensible way.

c. Ref 32 presents evidence for kinesin-based transport of IP3R in ER-derived vesicles, which is distinct from the transport of reticular ER IP3R reported here. The authors should comment on this difference and clearly delineate the evidence that IP3R transport in HeLa cell system is in the reticular ER rather than vesicles.

4. Selective activation of immobile IP3R near ER-PM junctions.

This is really the crux of the paper, and is based on two observations: practically all puffs occur at immobile IP3R clusters near the PM, and 84% of these clusters are associated with STIM1 puncta. The data in support of these key observations needs to be strengthened to justify the model for local crosstalk between IP3R and STIM/Orai proposed in Fig. 7.

a. On p. 9, the authors make very strong statements that Ca puffs are restricted to peripheral immobile IP3R clusters. This seems to be based on spinning disk confocal measurements at the base of the cell and in “a plane across the middle of the cell.” This is a key conclusion and the experimental support needs to be more completely described. Also, it is not clear whether caged IP3-induced puffs are also restricted to immobile puncta near the PM (in addition to histamine-induced puffs). I.e., are immobile puncta deeper in the cell unresponsive to IP3, rather than simply being farther from the source of IP3?

b. Ca²⁺ release is most prominent from larger clusters, which are often immobile, and it does seem that they are centered on the examples shown in Fig. 6, but in the movies it seems that there are other puffs that don't cleanly localize to a cluster (videos 6 and 7). What proportion of

puffs do not fall cleanly on top of an immobile IP3R cluster, and could these result from puncta located deeper within the cell?

c. The argument against puff sites having a larger number of IP3R seems weak to me. Basically, the intensity distribution of puncta producing puffs vs. total puncta (6f) is hard to interpret, because intensity reflects not only the number of IP3R1 subunits and the number of tetramers, but also proximity to the PM. The puff sites tend to be brighter, but is that because they have more IP3R1 subunits, more receptors or because they are localized near ER-PM junctions? One cannot tell from these data. As I note above, the unusually bright puncta (detailed in Fig. 1 and 2) do not seem very relevant to this paper, since they account for such a small fraction of the puff sites.

d. On p. 10, Fig. 7, the authors need to show how they calculate that 84% of immobile puncta are associated with STIM1. It is not possible to glean this from Fig. 7a-d, and it looks like many puncta are actually not near STIM1. It would help to show a panel with the outlines of the STIM1 and IP3R puncta identified by the Fiji plugins.

e. On p. 10, the authors state “The sites to which STIM1 translocated were selectively associated with immobile, rather than mobile, IP3Rs (Fig. 7e)”. This is not well supported by the data in Fig. 7e. The graph shows that STIM1 increases significantly near mobile IP3R, and the images in 7f show multiple IP3R puncta in close association with STIM1 (I assume that the clusters not labeled with the arrowhead are mobile). In addition, it is unclear why STIM1 fluorescence increases in ROIs of IP3R puncta, if the two are not coincident. The conclusion here seems overstated.

Minor points

1. In Fig. 1a, presentation of ER-tracker data is redundant, and the point of colocalization of ER and IP3R is much more convincingly made from the mCh-ER data. I suggest deleting 1a to conserve space.

2. Figs. 7 and 8 are largely redundant. Fig. 8 could be moved to supplemental.

Reviewer #2 (Remarks to the Author):

Review Comments

In this manuscript, the authors tried to reveal the relation between Ca²⁺ signals initiation site and the mobility of IP3Rs by using gene-editing technique to tag endogenous IP3R1s with mEGFP. The findings that Ca²⁺ signals initiate at immobile IP3R adjacent to the ER-PM junction where STIM evokes Ca²⁺ entry seems to be interesting for readers but is not enough for publication. As is mentioned below the present manuscript has many concerns.

Major comments:

(1) In this manuscript, the authors argue that “Most IP3R are mobile, moved by diffusion and motor proteins. The immobile IP3Rs do not mix with mobile IP3Rs”. In this kind of situation, what is the mechanism how immobile IP3Rs move to PM? The authors should show an experimental data to explain about the transfer mechanism of “immobile” IP3Rs. In addition, how does immobile IP3R clusters formation was initiated? a) Mobile IP3Rs trapped onto the existent ER-PM structure? Or b) Immobile IP3R cluster initiates the formation of the active spot for ER-PM junction? Please show any experimental data to show the relationship between immobile IP3R cluster formation and ER-PM junction. This is a very important point to confirm the hypothesis proposed by the authors.

(2) We sometimes observe punctate autofluorescence, especially in dim samples, as shown in Figure 1f. Therefore, to confirm that fluorescent signals were only derived from EGFP-IP3R1, the authors should show data as mentioned below: (1) Fluorescent image of wild type HeLa cell as a negative control. (2) Immunofluorescent staining data of EGFP-IP3R1 cells with anti-GFP or anti-IP3R1. (3) Whole-cell images of Figure 1a and 1b.

(3) In Supplementary Figure S2b, the authors mentioned that unaltered expression of IP3R2 and IP3R3. However, the expression levels of IP3R2 and IP3R3 decreased to 83-89% and 83-92%, respectively. There seems to be statistically significant differences between wild type cells and EGFP-IP3R1 cells. These altered expressions of IP3R2 and IP3R3 may affect the distribution of IP3R1. The authors should not ignore these differences.

(4) Based on Supplementary Figure S2h and 2I, the authors argue that EGFP-IP3R1 was functional. There is no proof for it. The authors should perform experiments as following: (see below). (1) In Supplementary Figure S2h and 2I, the authors used HeLa cells expressing both non-tagged IP3R1 and EGFP-IP3R1 (~55% EGFP-IP3R1). In this situation, it is difficult to observe the Ca²⁺ release activity of EGFP-IP3R1. Therefore, the authors should use the cells for siRNA analysis expressed only EGFP-IP3R1. (2) To test the Ca²⁺ release activity of EGFP-IP3R1, the authors should also use wild type HeLa cells as a control. (3) There are some concerns about the concentration of histamine (100 μM) used for these experiments. High-dose stimulations is well known to hinder the small, but significant, differences of Ca²⁺ responses. Therefore, the authors should perform Ca²⁺ imaging with 1 μM (or 5 μM) of histamine that is

enough to induce IP3-induced Ca²⁺ release from ER.

(5) The criteria of “immobile” are not clear and obscure. In Fig 2, there seems to be no descriptions about criteria. In Fig 3, 6, 7 and 8, the puncta that were immobile for 30 s were identified as “immobile”. In contrast, the definition of immobile is 10 s in Fig 4 and 8. The different criteria misidentify the different population as if they belong to the same one. Therefore, the authors should confirm the criteria of immobile throughout the manuscript.

(6) Related to the comments above, the authors identified immobile puncta by overlaying two successive frames (10 or 30 s apart) using pseudocolors for each (green and magenta), such that immobile puncta appear white in the overlay. However, in this kind of situation, there is a possibility that mobile puncta were identified as “immobile” puncta because mobile puncta are likely to move to the site where other mobile puncta were exist. This kind of strategy used in the experiment leads to misidentification. Therefore, the authors should use more straight forward scientific methods for the identification because classification of puncta is a key point of this work.

(7) At p5, the authors argued that mobile puncta did not interact with immobile punctum based on Video 2. However, it is very difficult to distinguish whether mobile puncta interact with immobile punctum or not. The authors must logically give reasons as to why the authors concluded that there is negligible exchange of IP3Rs between mobile and immobile puncta.

(8) At p7, the authors mentioned that “but it had no effect on the mobility of EGFP-IP3R1 within ER”. The authors should show the statistical data about this.

(9) In Supplementary Fig. S6c, to confirm the interaction between IP3R1 and dynein, the authors should also show the results of WB with anti-dynein after IP with anti-GFP.

(10) In Supplementary Fig. S6g, the size of accumulation of KIF5C-FKBP and Lyn11-FRB in GFP-ER expressing cell is relatively smaller than EGFP-IP3R1 HeLa cell. Thus, from these results, is difficult to conclude that kinesin-1 convey EGFP-IP3R1, but not GFP-ER, to PM. Therefore, the authors should show another pictures of GFP-ER expressing cells that have large accumulation of KIF5C-FKBP and Lyn11-FRB same as EGFP-IP3R1 HeLa.

(11) In Figure.7, the authors showed the co-localization of EGFP-IP3R1 and STIM1. However, it is not clear whether overexpressed STIM1 are the same localization as the endogenous STIM1 or not, since overexpression may alter its distribution. Therefore, the authors at least should show the distribution of endogenous STIM1. In addition, to confirm the translocations of STIM1 to ER-PM contact site where immobile IP3R1 exist contribute to SOCE, the authors should also access whether Orai also exist on the same site or not. In abstract, the authors concluded that

“IP3Rs tethered close to ER-plasma membrane junctions are licensed to respond and optimally placed to respond to endogenous IP3 and regulate Ca²⁺ entry through local depletion of Ca²⁺ stores.” However, it is difficult to understand the logic, since there are no evidences directly indicate the regulation of local Ca²⁺ entry by immobile IP3R. The authors should show experimental data to indicate directly link between the immobile IP3Rs cluster formation and local Ca²⁺ entry.

Minor comments:

- (1) No citations of TALENs. References should be added.
- (2) The genome sequences of gene-edited sites should be shown.
- (3) The WB data in Figure 1e was indistinct. Please exchange the image for more distinct one.
- (4) The authors mentioned about “stepwise photobleaching” at p4. The ref.23 (Ulbrich et al., Nat Methods, 2007) should be cited after this word, because Ulbrich et al firstly reported the stepwise photobleaching.
- (5) Cite the references about the description “We present estimates ~ by diffusion alone” at p5.
- (6) In Figure 5g, the authors should describe the definitions of Immobile/sub-diffusive, Confined, Diffusive, and Directed.
- (7) Because it is difficult to compare the Figure 4d and 4e, TIRFM and STORM images in Figure 4d and 4e should be shown at the same scale.
- (8) At p11, the authors mentioned that “Our results are consistent with evidence that kinesin mediates movements of IP3Rs along the ER of hippocampal neurons (Ref 32)”. However, Bannai et al, the authors of ref 32, showed that kinesin mediates movements of the vesicular sub-compartments of ER that contain IP3R, not IP3R itself. Therefore, the description inadequate and should be changed.
- (9) At p37, the authors cited the Almonacid et al (ref 48) related to the Fiji TrackMate plugin. However, original article about this technique is Jaqaman et al. “Robust single-particle tracking in live-cell time-lapse sequences. Nat Methods (2008) vol.5 (8) pp695-702. Therefore, the authors should cite the article.

Reviewer #3 (Remarks to the Author):

Technical comments:

"We define an immobile punctum as one for which there is no detectable movement in frames separated by 30 s. Immobile puncta were identified by overlaying TIRFM images of EGFP fluorescence collected 30-s apart; the images were pseudocoloured (green for $t = 0$, magenta for $t = 30$ s) such that overlaid images (i.e. immobile puncta) appeared white (see Fig. 3b-d)."

This is a qualitative measure, I wonder why the authors didn't instead use centroid finding of the punctae and a threshold of displacement/time based on the motion of the ER.

The FRAP parameters used for the puncta turnover (rectangular, 30×7 μm) were very different from those used for measuring overall diffusion coefficients (round ROI, ~ 2 μm radius). This should be explained: what is being optimised here?

"If IP3Rs were tightly packed within puncta, we might expect the fluorescence intensity of a punctum to correlate with its area, but there was no such correlation (Fig. 4a)." This doesn't make much sense to me, since the puncta were stated to contain on average 2 tetramers, which should be diffraction-limited. If the punctae are diffraction limited, they should all appear to be the size of a diffraction-limited spot.

Related to this, I don't understand the statement "We conclude that IP3Rs are corralled within mobile and immobile puncta by mechanisms that do not require direct contact between IP3Rs." Yes, if you are doing diffraction-limited imaging, the best you can do is to track diffraction-limited spots. And at any given time, a spot may be composed of more than one complex. But this doesn't mean that the complexes within that spot are all behaving the same way. This statement seems false or unproven.

"If all mobile IP3Rs moved randomly by diffusion, the relationship between time and mean squared displacement (MSD) should be linear." This is only true if you are measuring objects moving in 2D or 3D. The ER has a complex topology -- this statement is simply not true. Of course, there still may be directed motion present, but the test described in this statement will not work for complex topologies which constrain trajectories as organelles can.

Our specific responses to the reviewers' comments are detailed below (wherein bold figure numbers refer to numbering in the revised manuscript):

Reviewer #1

In this study by Taylor and colleagues, gene editing was used to label endogenous IP₃Rs and describe their organization, mobility, and function in HeLa cells. The authors present evidence that IP₃Rs are clustered, move by diffusion and by microtubule-based motors, and that the only receptors that respond to IP₃ are an immobile fraction located mostly next to ER-plasma membrane junctions. This last result is probably the most novel and implies that IP₃Rs may cause local ER Ca²⁺ depletion that controls nearby store-operated Ca²⁺ channels. The generation and validation of the GFP-IP₃R cells was carefully and thoroughly done. Using this approach, the authors confirm several results previously observed for heterologously expressed IP₃Rs, such as receptor mobility, clustering and preferred activity of immobile receptors. The two main new findings of the paper are that IP₃Rs move in a directed fashion, with contributions from kinesin and dynein, and that immobile receptors located near STIM1 and ER-PM junctions are "licensed" to respond to IP₃. A number of automated image analysis algorithms are used throughout the paper, but a major problem is that these tools are not validated for the system under study, which makes it difficult or impossible to judge the results. In many places, the conclusions are overstated based on the data that are shown, while in others too much space is devoted to data and analysis that do not add significantly to the paper. Most importantly, I think more data are needed to support the conclusion that IP₃ and PI-linked receptors selectively activate immobile IP₃Rs adjacent to STIM1 and ER-PM junctions.

In responding to specific comments, we provide comprehensive validation of the algorithms used for image analysis (points 1a,c, 3a,b); revise and further verify our analysis of single-step photobleaching (point 1b-d); abbreviate presentation of the numbers of IP₃Rs within puncta (point 1a); and present additional data that substantially reinforce our key conclusions, notably in showing that IP₃ selectively activates immobile IP₃Rs (point 4a,b) and that STIM/Orai accumulate alongside immobile IP₃R puncta (point 4c).

Major comments

1. Number of IP₃Rs in clusters

Fig. 1 does show that IP₃Rs are associated with the ER, that they occur in clusters of varying intensities, and that the 3 isoforms combine. Although these things are mostly already known from other studies, these data do validate the endogenous labeling. However, I did not find the quantitative analysis to be informative:

a. I don't see the point of the intensity quantitation in 1c and Supp Fig. 4a. The intensity distribution is a product of several factors, including number of receptors, number of GFPs per receptor, and position in the TIRF field. Given these complications (which the authors acknowledge), the presentation of these numerous histograms seems descriptive without any clear interpretation. In addition, later in the paper it becomes clear that the brightest puncta highlighted here do not serve any particular function. So what is the point of describing them in such detail?

We agree that the point was laboured. We do need to introduce the distribution of fluorescence intensities at some point since they provide an assessment of whether particles amenable to specific analyses are an unbiased sampling of the population. We have removed

text describing the distribution of fluorescence intensities (p4), removed Fig. 1h and Supplementary Fig. 4a, and moved Fig. 1c to **Supplementary Fig. S5a**. In responding to point 1b, we revised our analysis of single-step photobleaching and streamlined presentation of the panels demonstrating that selection of particles for this analysis was unbiased (**Supplementary Fig. S5b,c**).

The reviewer correctly states that the fluorescence intensity distribution (**Supplementary Fig. S5a**) depends on both the location of puncta within the TIRF field and the number of fluorophores within the puncta. We added text to make this clear (p10-11).

b. The number of tetrameric IP₃Rs per punctum is questionable. The analysis of step photobleaching grossly underestimates the number of steps. It is well known that because of the flickery nature of GFP fluorescence, it becomes increasingly difficult to resolve steps in particles having more than 3-4 GFPs. Thus, the mean and the distribution shown in Fig. 1h, i are not meaningful. In fact, when the peak fluorescence is divided by the mean amplitude of a single photobleaching event, the mean number is almost twice what the step counting indicates (Supp. Fig. 4). Representative automated fits to traces with 5-22 steps should be shown, as well as the percentage of puncta that could not be quantitated in this way, considering that the reliability of such fits is questionable.

This valuable comment prompted us to look more closely at the automated analysis of stepwise photobleaching. Direct comparison of the same puncta using automated and manual analyses (where we use the amplitude of the final bleaching step and the initial fluorescence uniquely for each punctum to estimate the number of bleaching steps) confirmed the reviewer's concern. The automated analysis systematically underestimates the quantal content of the brightest puncta. We therefore manually analyzed more puncta and confirmed that there was no bias in our selection of puncta used for the analyses (**Supplementary Fig. S5**). The legend states the number of puncta selected and the number amenable to manual analysis. The data derived from automated analysis has been replaced by data derived from manual analysis (8.4 IP₃Rs/punctum) (**Fig. 1f**). We also show further examples of traces used for photobleaching analysis (**Fig. 1e, Supplementary Fig. S8**). The methods have been modified to describe the revised analysis and to briefly mention the limitations of automated analysis (p29). We thank the reviewer for this particularly helpful insight.

c. The automatic particle detection method is critical in this paper (for IP₃R clusters and STIM1 puncta) and must be validated by showing examples of what it does (and does not) classify as a punctum, and that it is detecting GFP and not punctate background fluorescence. This will greatly affect interpretations throughout the paper. Without such validation, the reader has to have blind faith that the algorithm is doing the right thing, which is not a given when signals are small and noisy.

We had validated the automated particle detection method (TrackMate), which we use only to detect EGFP-IP₃R1 puncta, but chose to omit it from the original manuscript to save space. A full validation of the TrackMate software was published¹ while our work was under review. This publication is cited (p28, para 2). We added a new figure (**Supplementary Fig. S4**) in which we show images demonstrating that TrackMate detects no puncta in wild-type HeLa cells, and that within EGFP-IP₃R1 HeLa cells it reliably detects all discernible puncta with no evident false-positives. We have added a new section to the methods (**Automated detection of fluorescent puncta**), within which the new figure is cited (p28).

d. In Fig. 2, step photobleaching data should be shown for bright vs dim puncta to support Fig. 2d, e. It is particularly important to state the percentage of puncta that could not be analyzed because of a lack of clearly defined steps. Because such puncta are more likely to have greater number of receptors, the method inherently underestimates the number. Because of these uncertainties, I do not think the GFP tagging method provides better quantitation of IP₃R numbers than the Ca²⁺ puff method used by Parker's group (ref. 30).

A new figure (**Supplementary Fig. S8**) shows examples of traces used for the manual photobleaching analyses of dim and bright puncta. Because the analyses are manual (we used manual analyses for these puncta in the original manuscript too), the inclusion criteria require only that we can resolve the final bleaching step, rather than every intervening step. The

numbers of puncta that passed this selection criterion (~80% for the comparison of dim and bright puncta) are provided in the legend to the new figure.

We disagree with the suggestion that Ian Parker's elegant analysis of Ca^{2+} release events provides a better measure of the number of IP_3Rs within a punctum. He estimates only the number of *active* IP_3Rs , whereas our results are the first attempt to quantify the *total* number of IP_3Rs . The distinction is important because we show that many EGFP- $\text{IP}_3\text{R1}$ puncta are functionally silent (**Fig. 6, Supplementary Fig. S6, S15, S17**) and would not be detected with Parker's methods. Inclusion of a new figure showing that neither histamine nor photolysis of caged IP_3 evokes Ca^{2+} puff activity away from the PM (**Supplementary Fig. S16**) reinforces this important point. Only by comparing *all* IP_3Rs with those that mediate Ca^{2+} release are we able to develop our important conclusion that only some IP_3Rs are licensed to respond.

2. The organization of IP_3R in puncta imply a corral for IP_3Rs

In Fig. 4, the STORM analysis does not clearly delineate single IP_3Rs . Please show a clear example of two IP_3Rs separated by up to ~200 nm as stated in the text. It is possible that an immobile punctum such as presented in 4d is actually composed of multiple independent clusters in which the IP_3R interact directly with each other, rather than being "corralled" by an unknown scaffold. Mobile puncta would be the best way to show movement of a collection of widely spaced IP_3Rs , but there is only one example that is convincing (bottom panel in 4e). Perhaps more examples like that can be presented, rather than ones that show only a very small punctum with one receptor.

We did not intend to imply that localization spots within our STORM images are individual IP_3Rs . The effective resolution of our STORM imaging is ~45 nm, within which at least four IP_3Rs could reside if tightly packed. We make this clearer in the revised text (p6, middle). The key point is that within a punctum some of the localization spots (arising from single or multiple IP_3Rs) are too far apart for them to be associated by direct interactions between IP_3Rs . The reviewer correctly states that mobile puncta provide the clearest way to show that several loosely associated IP_3Rs move as a corralled cohort. We now include additional examples of mobile puncta with clusters of IP_3Rs that are well-separated (**Fig. 4c**).

3. Analysis of IP_3R mobility

a. In Fig. 5, the classification of particle trajectories into diffusive, immobile, or directional classes is problematic. It is well known that the stochastic nature of diffusion can create a bewildering variety of MSDs over short time periods, which can appear to be supra or sublinear (e.g., see Saxton, Biophys J 64:6, 1766-80). The method used for automatic sorting into these categories needs to be validated, e.g. by analyzing computational models of particles with known modes of mobility. How would particles alternating between diffusion and active transport be treated by the algorithm? In addition, more information can be recovered from the data. The authors should show the average MSD vs time for the population of particles characterized as diffusive; what D does it predict? For those that are directed, the MSD vs time should allow an estimate of the speed of the motor, and is this roughly consistent with kinesin and/or dynein turnover rates?

For our classification analyses, we include *only* particles that could be tracked for long periods (>30 frames, i.e. >3 s) to avoid problems associated with analysis of very short trajectories². For comparison, in Parker's MSD analyses of IP_3R mobility³ single-particles were tracked for 4-40 frames (0.6-6 s). We added a comment on these issues and the relevant citation² to methods (p28-29).

While our work was under review, a comprehensive validation of TraJClassifier software was published⁴, which the authors had kindly provided ahead of publication. This contains computational models and demonstrates reliable categorization of particle trajectories into immobile, confined, diffusive and directed. This publication is now cited in the methods (p29 and legend to **Supplementary Fig. S10**).

Where particles switch between mobility modes (eg, from diffusion to active), TraJClassifier splits the single trajectory into two sub-trajectories (each with its own classification). Hence, by comparing the number of particles tracked (4359 in total) to the number of classified trajectories (4625), we can identify the fraction ($133/4492 = 3\%$) of particles that changed their form of motion during the recording. The comparison reveals that

only a very small number of particles (~3%) switch between mobility modes during the >3-s recording. The methods have been expanded to elaborate how TraJClassifier deals with changes in motion (p29) and text has been added (p7, middle) to note that very few particles change their motion during the recording.

We now show examples of MSD-time curves for each category of motion and the associated curve-fits (**Supplementary Fig. 10**). We have, as the reviewer helpfully suggested, used the MSD-time plots for diffusive particles to calculate D . The value obtained ($D = 0.0308 \pm 0.002 \mu\text{m}^2/\text{s}$) is reassuringly similar to that estimated by FRAP analyses ($D = 0.016 \mu\text{m}^2/\text{s}$). This improved estimate of D is now provided in the main text (p5, middle). We use this value of D to estimate the distance that diffusing puncta move between closely spaced recordings of Ca^{2+} signals and IP₃R locations (p10, middle; legend to **Supplementary Fig. S10**).

We thank the reviewer for suggesting that we also determine the speed of the particles categorised as moving directionally. This analysis is now presented in **Supplementary Figure S10,d,h**. From the raw data (80 puncta), the average speed is $0.324 \pm 0.040 \mu\text{m}/\text{s}$. This compares favourably with measured rates of particles moved by single dynein and kinesin motors^{5,6}. This new analysis is discussed in revised text (p8, bottom).

b. The evidence for motors in IP₃R transport is good, but the analysis in Supp. Fig. 6 is somewhat confusing. I do not know why the number of directional trajectories should be reduced if one type of motor is inhibited; rather, I would think the direction or velocity would be affected. Rather than plotting the number of directional trajectories, which is subject to the accuracy of the TrackMate algorithm, why not simply plot the MSD vs time for the various motor treatments? This would show the contribution of motors in a more readily comprehensible way.

We have substantially revised our presentation of the evidence relating to inhibition of microtubule motors. The most important new data show that simultaneous inhibition of both kinesin and dynein motors almost abolishes directional trajectories (**Supplementary Fig. S12f**), while modestly (but significantly) reducing the speed of the few remaining puncta that move directionally (**Supplementary Fig. S12g**). These results, we suggest, validate our use of the number of directional trajectories as a reporter of the requirement for motors for directional movement. We suggest that this presentation is more informative than an MSD-time plot, which is dominated by the behaviour of diffusing puncta.

In addition, we have now quantitatively analysed the effects of nocodazole, which we had previously reported as having no dramatic effect on the overall mobility of IP₃Rs as the ER retracted from the PM. By isolating only the directional trajectories for analysis, it is now clear that disruption of microtubules with nocodazole almost abolishes directional movement of IP₃R puncta without affecting the speed of the few remaining puncta that move directionally or D for those moving diffusively (**Supplementary Fig S11a-e**). The text has been revised to present the new analyses (p7, bottom). These effects of nocodazole might be consistent with directional movement occurring via either attachment of IP₃Rs to microtubules through end-binding (EB) proteins⁷ or loss of the tracks along which IP₃Rs are conveyed by microtubule motors. We therefore compared the distribution of mCh-EB3 with EGFP-IP₃R1 and confirmed that there was no association between directionally moving puncta and EB3. The new data, which reinforce our evidence attesting to the importance of microtubule motors, are presented in **Supplementary Fig. 11f-h, Video 4** and in revised text (p7-8).

c. Ref 32 presents evidence for kinesin-based transport of IP₃R in ER-derived vesicles, which is distinct from the transport of reticular ER IP₃R reported here. The authors should comment on this difference and clearly delineate the evidence that IP₃R transport in HeLa cell system is in the reticular ER rather than vesicles.

We provide new evidence that directional movement of IP₃R puncta is unlikely to be mediated by vesicular structures in HeLa cells (p8-9). Data showing that mobile IP₃R puncta are not associated with concomitant movement of an ER luminal protein and that directionally moving IP₃Rs follow trajectories that exactly overlay those of diffusing IP₃Rs are shown in **Video 6** and **Supplementary Fig. S14**.

4. Selective activation of immobile IP₃R near ER-PM junctions

This is really the crux of the paper, and is based on two observations: practically all puffs occur at immobile IP₃R clusters near the PM, and 84% of these clusters are associated with STIM1 puncta. The data in support of these key observations needs to be strengthened to justify the model for local crosstalk between IP₃R and STIM/Orai proposed in Fig. 7. We provide substantial additional data in support of both key conclusions, namely that only near-PM immobile IP₃Rs are licensed to respond (points 4a,b) and that immobile IP₃R clusters are at sites where STIM and Orai will interact (points 4d,e).

a. On p. 9, the authors make very strong statements that Ca puffs are restricted to peripheral immobile IP₃R clusters. This seems to be based on spinning disk confocal measurements at the base of the cell and in “a plane across the middle of the cell.” This is a key conclusion and the experimental support needs to be more completely described. Also, it is not clear whether caged IP₃-induced puffs are also restricted to immobile puncta near the PM (in addition to histamine-induced puffs). I.e., are immobile puncta deeper in the cell unresponsive to IP₃, rather than simply being farther from the source of IP₃?

This is an important point. We now present the original spinning disc confocal data with histamine more comprehensively (**Supplementary Fig. S16c,d**). In addition, we have performed similar experiments with caged-IP₃, which confirm that almost all Ca²⁺ puffs occur near the PM (**Supplementary Fig. S16a,b**). The fuller presentation allows us to show that even when Ca²⁺ puffs do occur in the middle of the cell, they are invariably at the periphery, close to the PM. The new observations are discussed in the text (p9, middle), which we have re-organised to segregate the evidence that puffs occur near the PM from the evidence that even within the near-PM IP₃Rs, only the immobile puncta respond (p9-10). Our conclusion that puffs occur predominantly near the basal PM concurs with a previous report, which is now cited⁸ (p9, middle). We hope that the revised text (including Discussion, p14, middle) now more clearly makes the point that to be ‘licensed’ to respond, IP₃Rs must be *both* near the PM *and* immobile. It is, therefore, a remarkably small subset of all IP₃Rs that are ‘licensed’.

b. Ca²⁺ release is most prominent from larger clusters, which are often immobile, and it does seem that they are centered on the examples shown in Fig. 6, but in the movies it seems that there are other puffs that don’t cleanly localize to a cluster (videos 6 and 7). What proportion of puffs do not fall cleanly on top of an immobile IP₃R cluster, and could these result from puncta located deeper within the cell?

Our quantitative analyses using 2 different methods to interleave identification of IP₃R puncta and Ca²⁺ signals (now fully described in legend to **Fig. 6f** and **Supplementary Fig 17e**) establish that ~80% of all Ca²⁺ puffs can be localized to an immobile IP₃R punctum (**Fig. 6f** and **Supplementary Fig 17e**). We also provide additional data using spinning disc confocal microscopy showing that Ca²⁺ puffs evoked by histamine or caged-IP₃ are very rare across the middle of the cell, and those that occur are located close to the PM (see point 4a).

We revised our presentation of **Video 8**, which shows Ca²⁺ puffs and the distribution of immobile IP₃R puncta, to show only a portion of the cell. The larger scale is better suited to showing the coincidence of Ca²⁺ puffs and immobile puncta. We anticipate that the more appropriately presented video will persuade the reviewer that our quantitative analyses are correct in concluding that Ca²⁺ puffs occur almost exclusively at immobile puncta.

c. The argument against puff sites having a larger number of IP₃R seems weak to me. Basically, the intensity distribution of puncta producing puffs vs. total puncta (6f) is hard to interpret, because intensity reflects not only the number of IP₃R1 subunits and the number of tetramers, but also proximity to the PM. The puff sites tend to be brighter, but is that because they have more IP₃R1 subunits, more receptors or because they are localized near ER-PM junctions? One cannot tell from these data. As I note above, the unusually bright puncta (detailed in Fig. 1 and 2) do not seem very relevant to this paper, since they account for such a small fraction of the puff sites.

We agree that we must be cautious in considering why bright puncta might disproportionately contribute to Ca²⁺ puffs. However, with ~40% of Ca²⁺ puffs initiating at sites with underlying fluorescence intensities within the 95th percentile of all puncta (**Fig. 6g**)

and evidence that large IP₃R clusters are more likely to respond⁹, we feel it is important to consider possible relationships between brightness and activity. The text (p10-11) has been modified to more explicitly acknowledge that a significant contribution to the brightness of the responsive puncta may come from them being closer to the PM (an argument that strengthens our conclusion that cluster size is not alone sufficient to explain the activity of immobile puncta).

d. On p. 10, Fig. 7, the authors need to show how they calculate that 84% of immobile puncta are associated with STIM1. It is not possible to glean this from Fig. 7a-d, and it looks like many puncta are actually not near STIM1. It would help to show a panel with the outlines of the STIM1 and IP₃R puncta identified by the Fiji plugins.

We expanded the legend to **Fig. 8d** to clearly explain the criteria used to establish quantitatively the percentage of immobile IP₃R puncta that abut STIM1 puncta. In addition we present additional data showing juxtaposition of immobile puncta with native STIM1 (**Fig. 7a-d**) and in the associated fluorescence intensity plots across cells we illustrate the distances between the centroids of STIM1 and immobile IP₃R1 puncta (**Fig. 7e**). Similar expanded data for the relationship between STIM1-mCherry/EGFP-IP₃R1 puncta are shown in **Fig. 8a-d** for cells stimulated with thapsigargin and in **Supplementary Fig. S18** for cells stimulated with histamine or thapsigargin.

To more quantitatively address the relationship between STIM1 and immobile IP₃R puncta, we measured distances between them. The results show that although STIM1 puncta are much more abundant than immobile IP₃R puncta, the distance between STIM1 puncta is larger than that between STIM1 and immobile IP₃R puncta (**Fig. 8f,g**). Furthermore the distribution of distances between STIM1 and mobile versus immobile IP₃R puncta is consistent with our suggestion that immobile IP₃R puncta are juxtaposed to STIM1 puncta but displaced from them (**Fig. 8h**). The new analyses (**Fig. 7e, 8d,h**) provide estimates of the average separation of STIM1 puncta and immobile IP₃R: $0.70 \pm 0.07 \mu\text{m}$, with very few coming within $0.3 \mu\text{m}$ (**Fig. 8h**). These additional observations – which substantially strengthen our conclusion that STIM1 puncta abut immobile IP₃R puncta – are described in revised text (p11, bottom).

e. On p. 10, the authors state “The sites to which STIM1 translocated were selectively associated with immobile, rather than mobile, IP₃Rs (Fig. 7e)”. This is not well supported by the data in Fig. 7e. The graph shows that STIM1 increases significantly near mobile IP₃R, and the images in 7f show multiple IP₃R puncta in close association with STIM1 (I assume that the clusters not labeled with the arrowhead are mobile). In addition, it is unclear why STIM1 fluorescence increases in ROIs of IP₃R puncta, if the two are not coincident. The conclusion here seems overstated.

We expanded our analysis and presentation of the data showing that immobile IP₃R puncta abut the junctions where STIM1 accumulates after store depletion (see point 4d). In the revised images (**Fig. 7d, 8b,c**) all immobile IP₃R puncta are marked, such that the other green puncta are mobile IP₃R puncta.

In new experiments, we expressed mCherry-MAPPER or massively over-expressed STIM1-mCherry: both caused massive expansion of the ER-PM junctions from which immobile EGFP-IP₃R puncta were excluded, but the mobile EGFP-IP₃R puncta apparently were not (**Supplementary Fig. S19f,g**). The new data are discussed in revised text (p12-13).

We had inadequately described our analysis of the changes in STIM1 fluorescence associated with ROI overlying mobile and immobile IP₃R puncta. Our approach involved indentifying the IP₃R puncta and then extending the ROI that enclosed it to an additional radius, such that changes in STIM1 fluorescence immediately adjacent to the IP₃R punctum would be detected. The method is described on p30 and in a revised legend to **Fig. 8d**.

Minor points

1. In Fig. 1a, presentation of ER-tracker data is redundant, and the point of colocalization of ER and IP₃R is much more convincingly made from the mCh-ER data. I suggest deleting 1a to conserve space.

We agree and have deleted the original Fig.1a.

2. Figs. 7 and 8 are largely redundant. Fig. 8 could be moved to supplemental.

These are the only main figures in which we present the relationship between STIM1 and IP₃Rs. Since the reviewer commented that the juxtaposition of immobile IP₃Rs and STIM1 is a significant conclusion of our work, we feel it is important to afford it appropriate attention in the main figures. However, in light of additional data relating to the relationship between IP₃Rs and SOCE (points 4d and e, Reviewer 2, point 11), we have substantially revised these figures. **Fig. 7-8**, which include new data, including analysis of native STIM1, now provide the key observations describing the relationships between STIM1 and immobile IP₃R puncta.

Reviewer #2

In this manuscript, the authors tried to reveal the relation between Ca²⁺ signals initiation site and the mobility of IP₃Rs by using gene-editing technique to tag endogenous IP₃R1s with mEGFP. The findings that Ca²⁺ signals initiate at immobile IP₃R adjacent to the ER-PM junction where STIM evokes Ca²⁺ entry seems to be interesting for readers but is not enough for publication. As is mentioned below the present manuscript has many concerns.

We have responded specifically to each of the concerns expressed.

Major comments

(1) In this manuscript, the authors argue that “Most IP₃R are mobile, moved by diffusion and motor proteins. The immobile IP₃Rs do not mix with mobile IP₃Rs”. In this kind of situation, what is the mechanism how immobile IP₃Rs move to PM? The authors should show an experimental data to explain about the transfer mechanism of “immobile” IP₃Rs. In addition, how does immobile IP₃R clusters formation was initiated? a) Mobile IP₃Rs trapped onto the existent ER-PM structure? Or b) Immobile IP₃R cluster initiates the formation of the active spot for ER-PM junction? Please show any experimental data to show the relationship between immobile IP₃R cluster formation and ER-PM junction. This is a very important point to confirm the hypothesis proposed by the authors.

The focus of the present work is to establish relationships between acutely mobile and immobile populations of IP₃Rs and IP₃-evoked Ca²⁺ signals, leading to our conclusion that it is immobile IP₃Rs that release Ca²⁺. It is, of course, interesting to ask how IP₃Rs acquire their different behaviours, but the details are beyond the scope of the present study.

We have, however, performed additional experiments using a non-perturbing STIM-derived construct, GFP-MAPPER, developed to quantify ER-PM junctions¹⁰. For these experiments, we used HEK cells because there is a cell line in which genes for all 3 IP₃R subtypes have been disrupted using CRISPR/Cas⁹¹¹. The results (**Supplementary Fig. S19a-e**) show that the number and size of the ER-PM junctions labelled with GFP-MAPPER are the same in HEK cells with and without IP₃Rs. These results, which are consistent with evidence that IP₃Rs are not directly required for SOCE in vertebrate cells¹², demonstrate that IP₃Rs are not required for assembly of ER-PM junctions.

We also include new data showing that when ER-PM junctions are expanded by expression of mCherry-MAPPER (a perturbing version of GFP-MAPPER) or by massive over-expression of mCherry-STIM1, the immobile EGFP-IP₃R1 puncta are pushed to the margins of the expanded junctions (**Video 9, Supplementary Fig. S19f,g**). The new observations, suggesting that junctions define the positions of immobile IP₃Rs rather than the other way round, are discussed in substantial new text (p12, bottom). New materials and methods have been added to the methods section.

(2) We sometimes observe punctate autofluorescence, especially in dim samples, as shown in Figure 1f. Therefore, to confirm that fluorescent signals were only derived from EGFP-IP₃R1, the authors should show data as mentioned below: (1) Fluorescent image of wild type HeLa cell as a negative control. (2) Immunofluorescent staining data of EGFP-IP₃R1 cells with anti-GFP or anti-IP₃R1. (3) Whole-cell images of Figure 1a and 1b.

We added a new figure (**Supplementary Fig. S4**) in which we provide evidence (including images of control and gene-edited HeLa cells) that TrackMate detects no puncta in wild-type HeLa cells, and that within EGFP-IP₃R1 HeLa cells it reliably detects all discernible puncta with no evident false-positives.

Our STORM images show overlays of GFP fluorescence with immunostaining for GFP (**Fig. 4**). We suggest that there would be no additional benefit from staining with anti-IP₃R1 antibodies (none of which are as good as the anti-GFP antibody). For the reviewer, we append an additional figure showing that imaging of EGFP-IP₃R1 directly or after immunostaining for GFP identify the same puncta (**Reviewer Fig. R1**). We see little benefit to including this figure in the manuscript.

Fig. 1a was removed at the request of Reviewer 1. The images shown in the original Fig. 1b (now **Fig. 1a**) were collected without matching images of the whole cell. We are not convinced that another image of EGFP-IP₃R and mCherryER would improve the manuscript, but we have included a TIRFM image of a whole cell (**Reviewer Fig. R2**) in the appended file for the reviewer. If the reviewer feels that this image should be included within the manuscript, we could add it to **Supplementary Fig. S2**.

(3) In Supplementary Figure S2b, the authors mentioned that unaltered expression of IP₃R2 and IP₃R3. However, the expression levels of IP₃R2 and IP₃R3 decreased to 83-89% and 83-92%, respectively. There seems to be statistically significant differences between wild type cells and EGFP-IP₃R1 cells. These altered expressions of IP₃R2 and IP₃R3 may affect the distribution of IP₃R1. The authors should not ignore these differences.

We agree with the need to provide statistical analysis of the relative expression levels of the IP₃R subtypes in wild-type and EGFP-IP₃R1 HeLa cells. This was not possible with the original data, some of which was from only 2 replicates. We performed further WB of the 2 cell lines and report the additional data (**Fig. 1c**), with the numbers confirming that IP₃R expression was unaltered by gene-editing (see legend to **Fig. 1c**).

(4) Based on Supplementary Figure S2h and 2I, the authors argue that EGFP-IP₃R1 was functional. There is no proof for it. The authors should perform experiments as following: (see below). (1) In Supplementary Figure S2h and 2I, the authors used HeLa cells expressing both non-tagged IP₃R1 and EGFP-IP₃R1 (~55% EGFP-IP₃R1). In this situation, it is difficult to observe the Ca²⁺ release activity of EGFP-IP₃R1. Therefore, the authors should use the cells for siRNA analysis expressed only EGFP-IP₃R1. (2) To test the Ca²⁺ release activity of EGFP-IP₃R1, the authors should also use wild type HeLa cells as a control. (3) There are some concerns about the concentration of histamine (100 μM) used for these experiments. High-dose stimulations is well known to hinder the small, but significant, differences of Ca²⁺ responses. Therefore, the authors should perform Ca²⁺ imaging with 1 μM (or 5 μM) of histamine that is enough to induce IP₃-induced Ca²⁺ release from ER.

We agree that there is a need to be entirely confident that the gene-edited EGFP-IP₃R1 can mediate IP₃-evoked Ca²⁺ release. We provide two additional sets of experiments to confirm this. As suggested by the reviewer, we report the effects of siRNA for EGFP on the Ca²⁺ signals evoked by a full range of histamine concentrations in HeLa cells with *all* IP₃R1s gene-edited to include EGFP. The results demonstrate that siRNA to EGFP (but not a control siRNA) selectively attenuates histamine-evoked Ca²⁺ signals (**Supplementary Fig. S2c-f**). We had originally included discussion of the HeLa cells heterozygously expressing EGFP-IP₃R1 only because those cells were used for the original siRNA experiments. There is no further need to include these cells since *all* experiments now use the HeLa cells with *all* IP₃R1 tagged with EGFP. We have accordingly revised **Supplementary Fig. S2** and associated text to remove discussion of cells with residual untagged IP₃R1.

In addition, we expressed EGFP-IP₃R1, with a linker sequence that matched that of the gene-edited protein, in HEK cells without endogenous IP₃Rs¹¹. Cells expressing this EGFP-IP₃R1 responded to carbachol with an increase in cytosolic [Ca²⁺], while there was no response from mock-transfected cells without IP₃Rs (**Supplementary Fig. S2g,h**).

These additional analyses confirm our conclusion that EGFP-IP₃R1 is functional (p4, top). The additional methods are described in the revised methods section.

(5) The criteria of “immobile” are not clear and obscure. In Fig 2, there seems to be no descriptions about criteria. In Fig 3, 6, 7 and 8, the puncta that were immobile for 30 s were identified as “immobile”. In contrast, the definition of immobile is 10 s in Fig 4 and 8. The different criteria misidentify the different population as if they belong to the same one. Therefore, the authors should confirm the criteria of immobile throughout the manuscript.

We used paired images collected at intervals (10 s or 30 s) that were long relative to the time course of movement of mobile IP₃Rs, pseudocoloured the 2 images, and then overlaid them to identify immobile puncta. Most analyses used an interval of 30 s between these overlaid images, but it was 10 s for the live-cell TIRF imaging before STORM to minimise EGFP illumination. We now show that our overlay protocol identifies the same immobile puncta when the interval between paired images is varied between 5 s and 60 s. The relevant methods sections have been amended to more clearly describe our analyses of immobile puncta (**Fluorescence recovery after photobleaching (FRAP)**, p25); **Stochastic optical reconstruction microscopy (STORM)**, p26). We conclude that although there is some minor variation in the criteria used to define immobile puncta, each succeeds in identifying the same puncta. A new figure fully explains the method and its validation (**Supplementary Fig. S7**). Figure legends (**Fig. 3b,c, 4a, 6a, 7c, 8a,b, Supplementary Fig. S7, S15, S17-19**) have all been expanded to clarify the means of identifying immobile puncta.

(6) Related to the comments above, the authors identified immobile puncta by overlaying two successive frames (10 or 30 s apart) using pseudocolors for each (green and magenta), such that immobile puncta appear white in the overlay. However, in this kind of situation, there is a possibility that mobile puncta were identified as “immobile” puncta because mobile puncta are likely to move to the site where other mobile puncta were exist. This kind of strategy used in the experiment leads to misidentification. Therefore, the authors should use more straight forward scientific methods for the identification because classification of puncta is a key point of this work.

We agree in principle, but in practise there is negligible false identification of immobile puncta through different mobile puncta coming to occupy the same site in the two frames. The new **Supplementary Fig. S7** demonstrates that our overlay protocol identifies the same immobile puncta when the interval between paired images varies between 5 s and 60 s. This would not occur if a significant number of the overlays were due to coincident localizations of two mobile puncta in the two images. We suggest that our straightforward overlay method provides a convenient and reliable means of identifying immobile puncta, without recourse to time-consuming tracking methods and the attendant risk of photobleaching. The issue is elaborated in the legend to **Supplementary Fig. S7**.

(7) At p5, the authors argued that mobile puncta did not interact with immobile punctum based on Video 2. However, it is very difficult to distinguish whether mobile puncta interact with immobile punctum or not. The authors must logically give reasons as to why the authors concluded that there is negligible exchange of IP₃Rs between mobile and immobile puncta.

Our most important evidence for a lack of significant exchange between immobile and mobile puncta is the FRAP data (**Fig. 3e**). Data in **Video 2** and a new **Fig. 3f** provides additional evidence for the relative independence of mobile and immobile puncta, as a mobile punctum seemingly passes unperturbed through an immobile one. These results, discussed in revised text (p5, bottom), indicate that there is no detectable exchange of fluorescence between the 2 puncta.

(8) At p7, the authors mentioned that “but it had no effect on the mobility of EGFP-IP₃R1 within ER”. The authors should show the statistical data about this.

We have now quantitatively analysed the effects of nocodazole, which we had previously reported as having no dramatic effect on the overall mobility of IP₃Rs as the ER retracted from the PM. By isolating only the directional trajectories for analysis, it is now clear that disruption of microtubules with nocodazole (**Supplementary Fig S11a**) almost abolished directional movement of IP₃R puncta without affecting the speed of the few remaining puncta that moved directionally (**Supplementary Fig S11c-e**). The text has been revised to present the new analyses (p7-8). These effects of nocodazole might be consistent with directional movement occurring via either attachment of IP₃Rs to microtubules through end-binding (EB) proteins⁷ or loss of the tracks along which IP₃Rs are conveyed by microtubule motors. We therefore compared the distribution of mCh-EB3 with EGFP-IP₃R1 and confirmed that there was no association between directionally moving puncta and EB3. The new data, which reinforce our evidence attesting to the importance of microtubule motors, are presented in **Supplementary Fig. 11f-h, Video 4** and in revised text (p7-8).

(9) In Supplementary Fig. S6c, to confirm the interaction between IP₃R1 and dynein, the authors should also show the results of WB with anti-dynein after IP with anti-GFP.

We have been unsuccessful with the reciprocal IP using anti-GFP antibodies to precipitate dynein. We suggest that functional evidence of an interaction between IP₃R and microtubule motors is ultimately more instructive than evidence from co-IP, and have therefore performed additional functional analyses addressing the role of dynein and kinesin. We have substantially expanded our evidence supporting the contributions of kinesin and dynein (including use of mycalolide, another inhibitor of dynein, **Supplementary Fig. S12**) to the directional movement of IP₃Rs. The additions are described at length in our responses to Reviewer 1, point 3.

(10) In Supplementary Fig. S6g, the size of accumulation of KIF5C-FKBP and Lyn11-FRB in GFP-ER expressing cell is relatively smaller than EGFP-IP₃R1 HeLa cell. Thus, from these results, is difficult to conclude that kinesin-1 convey EGFP-IP₃R1, but not GFP-ER, to PM. Therefore, the authors should show another pictures of GFP-ER expressing cells that have large accumulation of KIF5C-FKBP and Lyn11-FRB same as EGFP-IP₃R1 HeLa.

We now include a more representative example of this experiment (**Supplementary Fig. S13c**).

(11) In Figure.7, the authors showed the co-localization of EGFP-IP₃R1 and STIM1. However, it is not clear whether overexpressed STIM1 are the same localization as the endogenous STIM1 or not, since overexpression may alter its distribution. Therefore, the authors at least should show the distribution of endogenous STIM1. In addition, to confirm the translocations of STIM1 to ER-PM contact site where immobile IP₃R1 exist contribute to SOCE, the authors should also access whether Orai also exist on the same site or not. In abstract, the authors concluded that “IP₃Rs tethered close to ER-plasma membrane junctions are licensed to respond and optimally placed to respond to endogenous IP₃ and regulate Ca²⁺ entry through local depletion of Ca²⁺ stores.” However, it is difficult to understand the logic, since there are no evidences directly indicate the regulation of local Ca²⁺ entry by immobile IP₃R. The authors should show experimental data to indicate directly link between the immobile IP₃Rs cluster formation and local Ca²⁺ entry.

We provide additional data showing immunostaining of HeLa EGFP-IP₃R1 cells with anti-STIM1 antibody. The results confirm that native STIM1 accumulates alongside immobile EGFP-IP₃R1 puncta (**Fig. 7a-e**). We have not yet found an Orai1 antibody that reliably stains native Orai1, but we do present data showing that CFP-STIM1 and mCherry-Orai1 form colocalized puncta after store-depletion (**Fig. 7f**), confirming thereby that the sites at which STIM1 accumulates are likely to represent sites of SOCE (p11, bottom)

We observed that Ca²⁺ release occurs at immobile EGFP-IP₃R1 puncta (**Fig. 6, Supplementary Fig. S15, S17**) and Ca²⁺ release through IP₃Rs causes formation of STIM1 puncta adjacent to immobile IP₃Rs (**Supplementary Fig. S18**) and activation of SOCE (not shown). We suggest, therefore, that local depletion of Ca²⁺ stores by IP₃Rs may locally regulate SOCE, but we have not yet demonstrated that directly and suggest that it lies beyond the immediate scope of the present work. We have accordingly tempered the text: the Abstract has been amended to remove reference to local depletion (last line), and the Discussion (p15-16) has been amended to make it clear that we are speculating that the arrangement of IP₃Rs and ER-PM junctions may allow local regulation of SOCE.

Minor comments:

(1) No citations of TALENs. References should be added.

We added a citation to TALENS¹³(p3, bottom).

(2) The genome sequences of gene-edited sites should be shown.

Supplementary Fig. S1a shows the genome sequences within exon 3 of IP₃R1 targeted by the TALENs. The sequences of monomeric EGFP (NCBI: WP_031943942.1 with A206K) and of human IP₃R1 (NCBI: NP_002213.5.) are readily available, and we define the sequence of the EGFP-IP₃R1 linker in **Supplementary Fig. S2a**. The methods (p19, bottom) describe how we sequenced genomic DNA from the edited locus of *itpr1* in 5 monoclonal cells lines.

We added text to this Methods section (p19 ,bottom) to state explicitly that the sequencing confirmed that the gene-editing produced the product illustrated in **Supplementary Fig. S2a**.

(3) *The WB data in Figure 1e was indistinct. Please exchange the image for more distinct one.*

In response to both this comment and the request to provide additional analysis of the relative expression of all IP₃R subtypes in the gene-edited cells (point 3), **Fig. 1c** has been replaced with WBs showing expression of all 3 IP₃R subtypes in wild-type and EGFP-IP₃R1 HeLa cells. The results confirm (p4, top) that gene-editing had no effect on expression of the 3 IP₃R subtypes.

(4) *The authors mentioned about “stepwise photobleaching” at p4. The ref.23 (Ulbrich et al., Nat Methods, 2007) should be cited after this word, because Ulbrich et al firstly reported the stepwise photobleaching.*

The reference has been added (p4, middle).

(5) *Cite the references about the description “We present estimates ~ by diffusion alone” at p5.*

The appropriate references have been added (p5, top).

(6) *In Figure 5g, the authors should describe the definitions of Immobile/sub-diffusive, Confined, Diffusive, and Directed.*

The terms are described rigorously in the newly cited publication that describes the development of TrajClassifier⁴ (p29). **Supplementary Fig. S10** has been introduced and provides a comprehensive description of the methods and validation.

(7) *Because it is difficult to compare the Figure 4d and 4e, TIRFM and STORM images in Figure 4d and 4e should be shown at the same scale.*

Within each panel, each of the 3 images is now presented at the same scale (**Fig. 4b,c**).

(8) *At p11, the authors mentioned that “Our results are consistent with evidence that kinesin mediates movements of IP₃Rs along the ER of hippocampal neurons (Ref 32)”. However, Bannai et al, the authors of ref 32, showed that kinesin mediates movements of the vesicular sub-compartments of ER that contain IP₃R, not IP₃R itself. Therefore, the description inadequate and should be changed.*

We have expanded our discussion of this issue with further references (p8, bottom; p14, top) and provided additional evidence to support our arguments that directional movement of IP₃Rs in HeLa cells is unlikely to be within vesicles distinct from ER (**Supplementary Fig. S14**). The changes are described fully in our responses to Reviewer 1, point 3c.

(9) *At p37, the authors cited the Almonacid et al (ref 48) related to the Fiji TrackMate plugin. However, original article about this technique is Jaqaman et al. “Robust single-particle tracking in live-cell time-lapse sequences. Nat Methods (2008) vol.5 (8) pp695-702. Therefore, the authors should cite the article.*

While our manuscript was under review, the complete description of the TrackMate algorithm was published¹. We have therefore replaced the original reference (Almonacid et al) with the newer reference.

Reviewer #3

"We define an immobile punctum as one for which there is no detectable movement in frames separated by 30 s. Immobile puncta were identified by overlaying TIRFM images of EGFP fluorescence collected 30-s apart; the images were pseudocoloured (green for t = 0, magenta for t = 30 s) such that overlaid images (i.e. immobile puncta) appeared white (see Fig. 3b-d)." This is a qualitative measure, I wonder why the authors didn't instead use centroid finding of the puncta and a threshold of displacement/time based on the motion of the ER.

Our method of defining immobile puncta is convenient, simple and reliable. It would be impracticable to apply a simple correction for ER displacement since the movement is different in different regions of the cell. In response to this and related comments from the

other reviewers, we added an additional figure (**Supplementary Fig. S7**) to demonstrate that the overlay method consistently identifies the same immobile puncta when the interval between overlaid images is varied between 5 and 60 s. This confirms the robustness of our method. Additional text has been added to the Methods (p25-26) to more clearly describe the analysis.

The FRAP parameters used for the puncta turnover (rectangular, 30 x 7 μm) were very different from those used for measuring overall diffusion coefficients (round ROI, $\sim 2 \mu\text{m}$ radius). This should be explained: what is being optimised here?

Additional text now describes the rationale behind the choice of FRAP regions more clearly (p26, middle). Small round ROIs ($\sim 2 \mu\text{m}$ radius) are consistent with previous studies¹⁴ and amenable to estimating D from published methods¹⁵. But small ROI are not optimal for simultaneously comparing recoveries of mobile and immobile puncta, since each would likely contain no more than a single punctum and movement of the ER during the long recovery period (1 h) might undermine analyses of small ROI. We therefore bleached large rectangular ROI (30 x 7 μm) for experiments designed to compare exchange of EGFP-IP₃Rs with mobile and immobile puncta (**Fig. 3**).

"If IP₃Rs were tightly packed within puncta, we might expect the fluorescence intensity of a punctum to correlate with its area, but there was no such correlation (Fig. 4a)." This doesn't make much sense to me, since the puncta were stated to contain on average 2 tetramers, which should be diffraction-limited. If the punctae are diffraction limited, they should all appear to be the size of a diffraction-limited spot.

We agree and have removed the original Fig. 4a. It was included only to provide an introduction to the STORM analyses, which are alone sufficient to develop our argument that within puncta not all IP₃Rs are closely associated. In response to another comment (Reviewer 1, point 2), presentation of the STORM images has been improved (**Fig. 4b,c**).

Related to this, I don't understand the statement "We conclude that IP₃Rs are corralled within mobile and immobile puncta by mechanisms that do not require direct contact between IP₃Rs." Yes, if you are doing diffraction-limited imaging, the best you can do is to track diffraction-limited spots. And at any given time, a spot may be composed of more than one complex. But this doesn't mean that the complexes within that spot are all behaving the same way. This statement seems false or unproven.

Our suggestion that IP₃Rs are loosely corralled within both mobile and immobile puncta derives from analysis of super-resolution STORM images. Our improved STORM images resolve localizations that are too far apart within a punctum for them to be bridged by direct interactions between IP₃Rs (**Fig. 4b,c**) (see response to reviewer 1, point 2). We now make it clear in the text that the resolution of these STORM images is insufficient to determine whether clusters of localizations comprise a single IP₃R or a small cluster of up to about four IP₃Rs (p6, middle). It is, however, clear that there are small clusters of localizations within a punctum that are substantially more than 20 nm apart and incapable, therefore, of being held together by direct interactions between IP₃Rs (p6, middle).

"If all mobile IP₃Rs moved randomly by diffusion, the relationship between time and mean squared displacement (MSD) should be linear." This is only true if you are measuring objects moving in 2D or 3D. The ER has a complex topology -- this statement is simply not true. Of course, there still may be directed motion present, but the test described in this statement will not work for complex topologies which constrain trajectories as organelles can.

We thank the reviewer for this insight. We agree that the complex topology of the ER may influence the MSD-time curves. However, the overwhelming effect of any topological constraint on diffusing particles would be to make the curve sub-linear, rather than supra-linear (**Fig. 5d,e**). We now refer to this issue in the text (p.7, top). In expanding our discussion of TrajClassifier (**Supplementary Fig. S10**), we provide examples of MSD-time plots for particles following each pattern of motion. These analyses demonstrate that diffusing particles within the ER do, in practise, have linear MSD-time plots (**Supplementary Fig S10a,e**), from which we estimate a diffusion coefficient (D) similar to that determined from FRAP analyses (p5, middle). Hence, while we agree with the reviewer's

key point, in practise it seems not to present substantial difficulties. Furthermore, we provide additional evidence to support our already strong evidence for a role for kinesin and dynein in directional movement of IP₃Rs (see response to Reviewer 1, point 3b).

1. Tinevez, J.Y. *et al.* TrackMate: An open and extensible platform for single-particle tracking. *Methods* **115**, 80-90 (2017).
2. Saxton, M.J. Lateral diffusion in an archipelago. Single-particle diffusion. *Biophys. J.* **64**, 1766-1780 (1993).
3. Smith, I.F., Swaminathan, D., Dickinson, G.D. & Parker, I. Single-molecule tracking of inositol trisphosphate receptors reveals different motilities and distributions. *Biophys. J.* **107**, 834-845 (2014).
4. Wagner, T., Kroll, A., Haramagatti, C.R., Lipinski, H.G. & Wiemann, M. Classification and segmentation of nanoparticle diffusion trajectories in cellular micro environments. *PLoS One* **12**, e0170165 (2017).
5. Friel, C.T. & Howard, J. Coupling of kinesin ATP turnover to translocation and microtubule regulation: one engine, many machines. *J. Muscle Res. Cell Motil.* **33**, 377-383 (2012).
6. Trokter, M., Mucke, N. & Surrey, T. Reconstitution of the human cytoplasmic dynein complex. *Proc. Natl. Acad. Sci. USA* **109**, 20895-20900 (2012).
7. Geyer, M. *et al.* Microtubule-associated protein EB3 regulates IP₃ receptor clustering and Ca²⁺ signaling in endothelial cells. *Cell Reports* **12**, 79-89 (2015).
8. Smith, I.F., Wiltgen, S.M. & Parker, I. Localization of puff sites adjacent to the plasma membrane: functional and spatial characterization of Ca²⁺ signaling in SH-SY5Y cells utilizing membrane-permeant caged IP₃. *Cell Calcium* **45**, 65-76 (2009).
9. Dickinson, G.D., Swaminathan, D. & Parker, I. The probability of triggering calcium puffs is linearly related to the number of inositol trisphosphate receptors in a cluster. *Biophys. J.* **102**, 1826-1836 (2012).
10. Chang, C.L. *et al.* Feedback regulation of receptor-induced Ca²⁺ signaling mediated by E-Syt1 and Nir2 at endoplasmic reticulum-plasma membrane junctions. *Cell Reports* **5**, 813-825 (2013).
11. Alzayady, K.J. *et al.* Defining the stoichiometry of inositol 1,4,5-trisphosphate binding required to initiate Ca²⁺ release. *Sci. Signal.* **9**, ra35 (2016).
12. Chakraborty, S. *et al.* Mutant IP₃ receptors attenuate store-operated Ca²⁺ entry by destabilizing STIM-Orai interactions in *Drosophila* neurons. *J. Cell Sci.* **129**, 3903-3910 (2016).
13. Gaj, T., Gersbach, C.A. & Barbas, C.F., 3rd ZFN, TALEN, and CRISPR/Cas-based methods for genome engineering. *Trends Biotechnol.* **31**, 397-405 (2013).
14. Pantazaka, E. & Taylor, C.W. Differential distribution, clustering and lateral diffusion of subtypes of inositol 1,4,5-trisphosphate receptor. *J. Biol. Chem.* **286**, 23378-23387 (2011).
15. Axelrod, D., Koppel, D.E., Schlessinger, J., Elson, E. & Webb, W.W. Mobility measurement by analysis of fluorescence photobleaching recovery kinetics. *Biophys. J.* **16**, 1055-1069 (1976).

We thank you and the reviewers for your careful attention to our work, and hope that with the changes we have made to the manuscript that it may now be acceptable for publication in *Nature Communications*.

Yours sincerely,

Colin W Taylor

Reviewer Fig. R1 | Imaging of GFP fluorescence and GFP immunostaining identify the same IP₃R puncta in EGFP-IP₃R1 HeLa cells. Typical TIRFM image of a fixed HeLa EGFP-IP₃R1 cell shows detection of EGFP directly and after immunostaining with an anti-GFP antibody. The overlay shows that the two methods identify the same puncta.

Reviewer Fig. R2 | TIRFM image of EGFP-IP₃R1 HeLa cell expressing mCherry-ER. (a) Complete cell. **(b)** Enlargement of boxed area.

Reviewers' comments:

Reviewer #1 (Remarks to the Author):

The authors have thoroughly addressed my concerns with the original version of the manuscript.

Reviewer #2 (Remarks to the Author):

Review Comments

In the rebuttal letter, the authors did not clearly answer my previous comments. As is mentioned below the present manuscript has many concerns.

Major comments

(1) As I already mentioned in the previous comments, the criteria of “immobile” were still not clear and obscure. The intervals of collected images were 10 seconds or 30 seconds. This meant that the authors could arbitrarily change the criteria of identifications, and that this identification method was far from scientific method. In addition, to answer my previous comments, the authors added Sup. Fig.7, however, in Sup. Fig.7, the fluorescent intensity and the shape of “immobile” puncta (white) seemed to be changed. That might be due to overlap of mobile puncta, as shown in Video 2, 3 and 6. These results strongly suggested that immobile puncta were not stable and that the method for identification of immobile puncta was not adequate and reliable. The new findings of this manuscript were that “Ca²⁺ signals are generated by a small population of immobile puncta, and immobile IP3Rs did not mix with mobile IP3Rs”, however, these conclusions were unreliable (See also comment (2) and (3)). Other results of this manuscript, such as movements of IP3Rs and the assembly of IP3Rs-STIM/Orai at ER-PM junctions, were already reported by other groups. Therefore the novelty of this manuscript was not enough and not suitable for the publication in Nature communication.

In addition, considered from the results of STORM data (Fig. 4d) showing that immobile puncta are larger and contain more IP3Rs than mobile puncta, “immobile puncta” used to be many “mobile puncta” to translocate to the sites adjacent to the PM and to form the large clusters. Therefore, I could not understand why the authors strongly denied the exchange between mobile and immobile puncta. Furthermore, in the authors’ imaging system, it was impossible to visualize single IP3Rs or small cluster that contained a small number of IP3Rs. Therefore, in this manuscript, the authors’ identification methods merely observed the “immobile structures” where a large amount of IP3Rs and other regulatory protein were easily accumulated. Thus, to begin with, the descriptions of mobile and immobile IP3Rs seemed to be inadequate.

As mentioned above, if the authors want to mention about mobile and immobile IP3Rs, not

“Structure”, the authors should, sincerely, use more straight forward and more scientific methods that are more reliable for the identification because classification of puncta is a key point of this work. The inappropriate method for identification seemed to decrease the importance of this study.

(2) In the rebuttal letter (7) (p9), the authors mentioned that “Our most important evidence for a lack of significant exchange between immobile and mobile puncta is the FRAP data (Fig.3e)”, however, the images of Fig.3d were dim and had very low signal-to-noise ratio. Therefore, it was difficult to confirm a lack of exchange between immobile and mobile puncta. The authors should have shown better representative data with higher signal-to-noise ratio. In addition, as I mentioned in previous review, based on Video2 and a new Fig.3f, it was completely difficult to distinguish whether mobile puncta interact with immobile puncta or not.

(3) On p9 – 10, the authors argued that “Ca²⁺ puffs occur at immobile puncta close to the PM”. However, as shown in Video 2, 3 and 6, mobile puncta overlapped with immobile puncta. Therefore, there was a possibility that mobile puncta that overlapped with immobile puncta induced the Ca²⁺ responses.

Minor comments

(1) Throughout the manuscript, the qualities of imaging data were relatively low.

(2) On p12, the authors mentioned that “Expression of mCherry-MAPPER in EGFP-IP3R1 HeLa cells causes massive expansion of ER-PM junctions, and this was accompanied by displacement of immobile EGFP-IP3R1 puncta to the margins of the enlarged junctions, without preventing mobile IP3Rs from passing through regions enriched in junctions”. However, I found it hard to understand this sentence. The authors should explain following questions.

(a) The authors should have explained about the function of non-perturbing and perturbing STIM that were used to construct GFP-MAPPER and mCherry-MAPPER

(b) The authors should have shown any scientific data to confirm that mCherry-MAPPER expand ER-PM junctions, instead of personal communication.

(c) I wonder how did the authors confirm that “without preventing mobile IP3Rs”.

In addition, it was very difficult to interpret the data in Sup. Fig19f-g and Video 9 due to low qualities of images.

(2) In this study, the authors used monomeric EGFP to tag endogenous IP3R1, therefore the authors should describe as “mEGFP-IP3R1” instead of EGFP-IP3R1 throughout the manuscript.

Reviewer #3 (Remarks to the Author):

Movie: 113284_1_video_2388289_lrf32 cell is significantly retracting (Video 5)

Response to rebuttal:

1. On the definition of immobile punctae: The authors misunderstood my suggestion to set a threshold based on ER motion. I wasn't advising a location-dependent correction, only a global threshold which takes into account that small displacements may occur due to ER motion, and still should be classified as immobile. I am not satisfied with the author's suggestion that a "by eye" assessment of whether a punctum appears "white" is preferable to the standard method for determining whether the centroid of a spot has displaced by more than a threshold amount. However, I don't wish to belabor this point, since sometimes the easy way is sufficient for the conclusions drawn. In fact, the "by eye" assessment is probably placing a threshold of ~10% displacement of the spot relative to its size, so 20-50 nm which is sensible.
2. The new description of the FRAP experiments makes sense.
3. The new STORM images show clearly small clusters within each punctum, supporting the authors' assessments.
4. The authors seem untroubled by the application of simple diffusion concepts to particles diffusing in the ER. They offer as evidence of its validity the consistency between FRAP and single particle tracking measurements. However, FRAP analysis also makes assumptions to output a diffusion coefficient, which are incorrect for a complex topology as well. Nevertheless, this doesn't interfere with the main conclusions related to directed motion and its connection to various motors.

The authors present a substantial work, with an impressive array of different methods applied to understand the motion and functionality of IP3 receptors.

Responses to reviewers

Reviewer #2

Major

(1) *As I already mentioned in the previous comments, the criteria of “immobile” were still not clear and obscure. The intervals of collected images were 10 seconds or 30 seconds. This meant that the authors could arbitrarily change the criteria of identifications, and that this identification method was far from scientific method. In addition, to answer my previous comments, the authors added Sup. Fig.7, however, in Sup. Fig.7, the fluorescent intensity and the shape of “immobile” puncta (white) seemed to be changed. That might be due to overlap of mobile puncta, as shown in Video 2, 3 and 6. These results strongly suggested that immobile puncta were not stable and that the method for identification of immobile puncta was not adequate and reliable. The new findings of this manuscript were that “Ca²⁺ signals are generated by a small population of immobile puncta, and immobile IP₃Rs did not mix with mobile IP₃Rs”, however, these conclusions were unreliable (see also comment (2) and (3)). Other results of this manuscript, such as movements of IP₃Rs and the assembly of IP₃Rs-STIM/Orai at ER-PM junctions, were already reported by other groups. Therefore the novelty of this manuscript was not enough and not suitable for the publication in Nature communication. In addition, considered from the results of STORM data (Fig. 4d) showing that immobile puncta are larger and contain more IP₃Rs than mobile puncta, “immobile puncta” used to be many “mobile puncta” to translocate to the sites adjacent to the PM and to form the large clusters. Therefore, I could not understand why the authors strongly denied the exchange between mobile and immobile puncta. Furthermore, in the authors’ imaging system, it was impossible to visualize single IP₃Rs or small cluster that contained a small number of IP₃Rs. Therefore, in this manuscript, the authors’ identification methods merely observed the “immobile structures” where a large amount of IP₃Rs and other regulatory protein were easily accumulated. Thus, to begin with, the descriptions of mobile and immobile IP₃Rs seemed to be inadequate. As mentioned above, if the authors want to mention about mobile and immobile IP₃Rs, not “Structure”, the authors should, sincerely, use more straight forward and more scientific methods that are more reliable for the identification because classification of puncta is a key point of this work. The inappropriate method for identification seemed to decrease the importance of this study.*

If, as the reviewer suggests, we mistakenly identify immobile puncta when two mobile puncta coincide at the same location, we would expect to identify different immobile puncta when the interval between overlaid images is varied. Results shown in **Supplementary Fig. S7a-d** show this does not occur: the same immobile puncta are identified whether the interval between overlays is 5, 10, 30 or 60 s. This analysis also confirms that our choice of interval is not ‘arbitrary’. Where we need to use different intervals (30 s usually, but 10 s for STORM to reduce photobleaching), we confirmed that each interval identifies the same immobile puncta (compare **Supplementary Fig. S7b with c**).

Reviewer #3 (point 1) had helpfully suggested an alternative approach, which for practical reasons is more difficult to implement. Nevertheless, and given the importance of the issue, we felt it important to compare that approach with ours. We compared immobile puncta identified by the overlay method with those identified by centroid displacements (using TrackMate and TraJClassifier to map the trajectories of puncta). The comparison, showing that *all* immobile puncta identified by overlaying images have trajectories classified as immobile/sub-diffusive or confined, is shown in a new panel (**Supplementary Fig. S7e**). These results demonstrate the coherence of the two approaches and concur with the suggestion of Reviewer #3 that our pragmatic approach is effective.

The reviewer suggests that the immobile puncta look different in each of the images shown in **Supplementary Fig. S7a-d**, suggesting that immobile puncta are unstable. We are puzzled by this statement since the immobile puncta seem remarkably consistent between the four sets of images. Nevertheless, we provide an additional analysis to allay the concern. A new panel shows that the fluorescence intensities of immobile puncta are stable across the intervals shown (**Supplementary Fig. S7f**).

It is not correct to state that we detect only large IP₃R clusters. We have analysed the distribution of IP₃Rs between puncta (**Fig. 1e,f, Supplementary Fig. S5, S8**) and shown that puncta contain from 1 to ~40 IP₃Rs (mean = 8.4, p4).

Our results show for the first time that: a) native IP₃R form loosely associated clusters, which we suggest may be the building blocks of IP₃-evoked Ca²⁺ signalling; b) IP₃R clusters differ in their mobility: some are immobile, while others move by diffusion or microtubule motors; c) only immobile clusters, and only those that are close to the PM evoke Ca²⁺ puffs, suggesting an additional level of IP₃R regulation that ‘licenses’ this small subset of IP₃R to respond; d) ‘licensed’ IP₃R are located adjacent to the machinery that regulates store-operated Ca²⁺ entry, suggesting a close spatial coupling of Ca²⁺ release and Ca²⁺ entry. These results, we suggest, will be of substantial interest within the field of Ca²⁺ signalling and beyond.

We provide evidence that Ca²⁺ puffs initiate at immobile puncta, which remain largely independent of mobile puncta over timescales of many minutes (**Fig. 3** and further supported by new data in **Supplementary Fig. S17f**). It will be important for future work to address, with measurements over much longer timeframes, how immobile IP₃R puncta are assembled and whether there is regulated exchange with mobile puncta (p14, para2). However, we suggest that these additional questions lie beyond the scope of the present study.

(2) In the rebuttal letter (7) (p9), the authors mentioned that “Our most important evidence for a lack of significant exchange between immobile and mobile puncta is the FRAP data (Fig.3e)”, however, the images of Fig.3d were dim and had very low signal-to-noise ratio. Therefore, it was difficult to confirm a lack of exchange between immobile and mobile puncta. The authors should have shown better representative data with higher signal-to-noise ratio. In addition, as I mentioned in previous review, based on Video2 and a new Fig.3f, it was completely difficult to distinguish whether mobile puncta interact with immobile puncta or not.

We believe that the images in **Fig. 3d** show clearly that after fluorescence bleaching, there is rapid recovery of mobile puncta (green and magenta), but not of immobile puncta (white). The images (**Fig. 3c,d**) were chosen to show results part-way through fluorescence recovery (15 min), but they must be collected with the same settings as the pre-bleach images (**Fig. 3b**); hence the recovering images are inevitably less bright than the initial ones. We have prepared a figure for Reviewer #2 (**Fig. R4**), from which it is clear from fluorescence profiles of the dimmest puncta collected at the first time interval used to monitor FRAP in **Fig. 3** (15 min; ie the dimmest images used) that we easily resolve fluorescent puncta from background.

The images in **Video 2** and **Fig. 3f** show that the positions and fluorescence intensities of immobile puncta are unchanged by transitory incursions from mobile puncta. Furthermore, we now provide additional analyses in **Supplementary Fig. S17f** (see response to point 3) showing that during a Ca²⁺ puff, mobile puncta do not invade immobile puncta.

(3) On p9-10, the authors argued that “Ca²⁺ puffs occur at immobile puncta close to the PM”. However, as shown in Video 2, 3 and 6, mobile puncta overlapped with immobile puncta. Therefore, there was a possibility that mobile puncta that overlapped with immobile puncta induced the Ca²⁺ responses.

The reviewer cites our observation of a mobile punctum seeming to pass through an immobile one as evidence that Ca²⁺ puffs may occur when mobile puncta collide with immobile puncta. We now provide additional evidence that this is not the case. For each of the Ca²⁺ puffs shown in **Supplementary Fig. S17**, we compared the fluorescence intensity of the underlying EGFP-IP₃R1 punctum for several seconds either side of the Ca²⁺ puff. The results demonstrate that the Ca²⁺ puffs evoked at immobile puncta are not associated with incursions from mobile puncta (**Supplementary Fig. S17f**). We thank the reviewer for prompting this additional analysis, which we suggest further strengthens our conclusion that immobile puncta initiate Ca²⁺ puffs.

The core concern behind each comment (1-3) relates to the Reviewer #2’s suggestion that we do not effectively identify immobile puncta. The issue is important, but we respectfully suggest that it is incorrect to assert that we do not reliably detect immobile puncta. Our method has been validated, and in response to constructive suggestions in previous reviews, its validation is clearly presented (**Supplementary Fig. S7, S17f**).

Minor

(1) *Throughout the manuscript, the qualities of imaging data were relatively low.*

We have, however, submitted high-resolution .tif images, rather than pdf files, of the figures in this revised submission.

(2) *On p12, the authors mentioned that “Expression of mCherry-MAPPER in EGFP-IP₃R1 HeLa cells causes massive expansion of ER-PM junctions, and this was accompanied by displacement of immobile EGFP-IP₃R1 puncta to the margins of the enlarged junctions, without preventing mobile IP₃Rs from passing through regions enriched in junctions”. However, I found it hard to understand this sentence. The authors should explain following questions.*

(a) *The authors should have explained about the function of non-perturbing and perturbing STIM that were used to construct GFP-MAPPER and mCherry-MAPPER*

We are sorry that the structures of the constructs were unclear in the original manuscript. We have amended the text (p12, para2) to make it clear that the STIM component is the same for both MAPPER constructs, only the tag (GFP or mCherry) differs.

(b) *The authors should have shown any scientific data to confirm that mCherry-MAPPER expand ER-PM junctions, instead of personal communication.*

Comparison of normal ER-PM junctions (diameter ~600 nm, **Supplementary Fig. S19d**) identified by immunostaining of native STIM1 (**Fig. 7a,b**), after low-level expression of mCh-STIM1 (**Fig. 8a-c, Supplementary Fig. S18**) or by GFP-MAPPER (**Supplementary Fig. S19b,c**) are so conspicuously smaller than those formed by mCh-MAPPER (**Supplementary Fig. S19f, Video 9**) that we suggest it is unnecessary to subject the comparison to further analysis.

(c) *I wonder how did the authors confirm that “without preventing mobile IP₃Rs”.* In addition, it was very difficult to interpret the data in *Sup. Fig19f-g and Video 9* due to low qualities of images.

We have modified presentation of these data. By superimposing a mask defining the area enclosed by an enlarged junction onto the overlaid images showing mobile and immobile IP₃R puncta, it is now clearer that the enclosed area is populated by mobile puncta (magenta and green) but not by immobile puncta (white) (**Supplementary Fig. S19f,g**). **Video 9** has been modified to include a third image showing mobile and immobile puncta superimposed on the image of the ER-PM junctions.

(3) *In this study, the authors used monomeric EGFP to tag endogenous IP₃R1, therefore the authors should describe as “mEGFP-IP₃R1” instead of EGFP-IP₃R1 throughout the manuscript.*

We now emphasise our use of monomeric EGFP in the Introduction (p3, para3), where we first define EGFP. We suggest that with EGFP defined here to mean monomeric EGFP there is no need to use mEGFP throughout the manuscript.

In addition, we have made the following minor revision:

Results (p4, bottom) and Discussion (p13, para 1): we have added text to more clearly state that the small clusters of IP₃Rs within puncta appear to be the elementary structural units of IP₃R signalling.

Reviewer #3

Movie: 113284_1_video_2388289_lrf32 cell is significantly retracting (Video 5).

We assume this comment relates to some retraction of the cell boundaries in **Video 5**, and that the reviewer may be concerned that such movement might, more generally, influence comparisons of IP₃R distributions and Ca²⁺ signals. Slow retraction of the PM would not affect the specific observations in **Video 5**, namely that trapping kinesin at the PM causes coincident trapping of EGFP-IP₃R1. We further suggest that such movement is unlikely to have affected other analyses:

- In **Video 5**, the cell was treated with rapamycin to re-distribute engineered kinesin, which may then have caused more movement than in a normal cell.

- The duration of the video (>17 min) is much longer than the timescales (30 s) used to assess IP₃R mobility and the coincidence of immobile IP₃R puncta and Ca²⁺ puffs. For the reviewer, we attach **Fig. R3** in which stills taken from **Video 5** show that even with rapamycin treatment the modest cell retraction takes much longer than 30 s.
- Our revised **Supplementary Fig. S7e,f** provides further specific evidence that cell movements do not perturb identification of immobile puncta (see response to point 1).

Response to rebuttal:

1. *On the definition of immobile punctae: The authors misunderstood my suggestion to set a threshold based on ER motion. I wasn't advising a location-dependent correction, only a global threshold which takes into account that small displacements may occur due to ER motion, and still should be classified as immobile. I am not satisfied with the author's suggestion that a "by eye" assessment of whether a punctum appears "white" is preferable to the standard method for determining whether the centroid of a spot has displaced by more than a threshold amount. However, I don't wish to belabor this point, since sometimes the easy way is sufficient for the conclusions drawn. In fact, the "by eye" assessment is probably placing a threshold of ~10% displacement of the spot relative to its size, so 20-50 nm which is sensible.*

The Reviewer acknowledges that our pragmatic approach (overlay of temporally separated images) to distinguish mobile and immobile puncta is effective. However, we recognize the importance of this issue, and accept that an alternative approach would have been to categorize immobile puncta on the basis of tracking centroid movements. Indeed, we use that approach to categorize the mobility of puncta (**Supplementary Fig. S10**). In a new **Supplementary Fig. S7e**, we show that *all* immobile puncta identified by image-overlay are also identified by centroid tracking (see also response to Reviewer #2, point 1).

2. *The new description of the FRAP experiments makes sense.*

3. *The new STORM images show clearly small clusters within each punctum, supporting the authors' assessments.*

We are pleased that these aspects are now clear.

4. *The authors seem untroubled by the application of simple diffusion concepts to particles diffusing in the ER. They offer as evidence of its validity the consistency between FRAP and single particle tracking measurements. However, FRAP analysis also makes assumptions to output a diffusion coefficient, which are incorrect for a complex topology as well. Nevertheless, this doesn't interfere with the main conclusions related to directed motion and its connection to various motors.*

We agree with these sentiments in that despite limitations in applying simple diffusion models to complex ER structures, the limitations do not imperil our conclusions. We note in passing that the puncta classified as diffusive by TraJClassifier (63% of all puncta, **Fig. 5g**) do display linear MSD plots (**Supplementary Fig. S10a,e**).

The authors present a substantial work, with an impressive array of different methods applied to understand the motion and functionality of IP₃ receptors.

We thank the reviewer for these comments.

Figure R3 | Response to Reviewer #3. Cell retraction is unlikely to contribute to acute analyses of the subcellular distribution of IP₃Rs or Ca²⁺ puffs. Images of EGFP-IP₃R1 taken from **Video 5** at the indicated times are shown as they are presented in the video (left) or with the contrast increased (in ImageJ) to clearly delineate cell boundaries (right). The boundary identified in the first image is superimposed on all subsequent images. The results demonstrate that after addition of rapamycin ($t = 0$, to trap engineered kinesin at the PM), there is some cell retraction (perhaps evoked by the tethering of kinesin to the PM), but it occurs over a period of several minutes, and is not evident within the shorter intervals (<30 s) used for our analyses of IP₃R distribution of Ca²⁺ puffs.

Figure R4 | Response to Reviewer #2. Reliability of the FRAP analyses shown in Fig. 3. (a,b) **Fig 3d** is reproduced on a larger scale (a) and shown with only one of the two overlaid images used to identify mobile and immobile puncta in the post-bleach analysis (b). Scale bars = 2 μm . For each of the 3 post-bleach images (b), fluorescence intensity profiles were analysed for bright (orange circles) and dim puncta (blue circles). (c,d) Results show fluorescence intensity profiles across each punctum for the indicated regions (1-3) for bright (c) and dim (d) puncta. The results demonstrate that even for the dimmest puncta identified at the first interval (15 min) used after bleaching (**Fig. 3**), fluorescent puncta can be readily distinguished from background fluorescence.